# Strategies for Improving Peptide Stability and Delivery

**DOI:** 10.3390/ph15101283

**Published:** 2022-10-19

**Authors:** Othman Al Musaimi, Lucia Lombardi, Daryl R. Williams, Fernando Albericio

**Affiliations:** 1Surfaces and Particle Engineering Laboratory, Department of Chemical Engineering, Imperial College London, London SW7 2AZ, UK; 2School of Chemistry and Physics, University of KwaZulu-Natal, Durban 4001, South Africa; 3CIBER-BBN, Networking Centre on Bioengineering, Biomaterials and Nanomedicine and Department of Organic Chemistry, University of Barcelona, 08028 Barcelona, Spain

**Keywords:** peptides, cell-penetrating peptides, stapled peptides, stitched peptides, hydrogel, self-assembled peptides

## Abstract

Peptides play an important role in many fields, including immunology, medical diagnostics, and drug discovery, due to their high specificity and positive safety profile. However, for their delivery as active pharmaceutical ingredients, delivery vectors, or diagnostic imaging molecules, they suffer from two serious shortcomings: their poor metabolic stability and short half-life. Major research efforts are being invested to tackle those drawbacks, where structural modifications and novel delivery tactics have been developed to boost their ability to reach their targets as fully functional species. The benefit of selected technologies for enhancing the resistance of peptides against enzymatic degradation pathways and maximizing their therapeutic impact are also reviewed. Special note of cell-penetrating peptides as delivery vectors, as well as stapled modified peptides, which have demonstrated superior stability from their parent peptides, are reported.

## 1. Introduction

With a molecular weight range from 500 to 5000 Da, peptides occupy an important position between large biological molecules (more than 5000 Da) and the small molecules (less than 500 Da) [1]. Peptides have demonstrated a remarkable performance in various fields of the medical research field. They are deployed in immunology [2], diagnosis [3], drug delivery [1,4,5], as well as biomaterials [6]. They also cover a wide spectrum as active pharmaceutical ingredients (API), carriers, linkers, among others [7]. Their medium size make them more suitable for functions that cannot be accomplished with either small or large molecules. For instance, while the large molecules are incapable of penetrating cells, the hydrophobic interaction is devoid with small molecules [8].

The number of novel therapeutic peptide approvals by the US Food and Drug Administration (USFDA) is increasing on a yearly basis, and currently there are approximately 120 peptides in the market [7,9,10]. About 140 peptide drugs are currently in clinical trials with more than 500 peptides in preclinical trials [7,11,12].

Modern peptide synthesis emerged after the brilliant invention of the Nobel laureate Merrifield in 1963 of solid-phase peptide synthesis (SPPS) [13]. More recently, the introduction of automatic peptide synthesizers, especially microwave-assisted versions, has also greatly facilitated the process whilst reducing the time and effort required for the synthetic process [14]. The implementation of new technologies such as flow chemistry [15] and on line real-time monitoring [16] can significantly optimize reagent usage as well as minimizing reaction time. These developments have facilitated the synthesis of peptides that were difficult to produce using conventional methods.

Despite the biological specificity and the promising safety profile of peptides, they suffer from poor proteolytic stability and short half-lives. These cause a loss of their secondary structure, and thus reduced activity versus time.

Synthetic modifications are one of the possible solutions to enhance the stability of peptides. For instance, swapping L-amino acids with their D-enantiomers provides peptides which are resistant to proteolytic degradation, thereby increasing the half-life [1,17]. However, if the amino acid swapping process compromises the new peptide’s biological activity, then the hydrocarbon stapling or retro inverso strategies are more favorable alternative synthesis tactics [18]. Improved formulations offer an alternate route to deliver the parent peptide without the need to modify it. Figure 1 shows all the strategies covered in this review for the sake of improving peptide stability and delivery.

In this review, we will be exploring this stability challenge in the framework of cell-penetrating peptides and their role in delivering peptides to their therapeutic targets. Furthermore, how the stapling strategy helped in achieving better stability, and hence the performance of peptides, will be highlighted. Finally, some intelligent delivery systems will be discussed as well.

## 2. Cell-Penetrating Peptides (CPPs)

CPPs are a novel class of peptides with a unique cell penetration ability and are also called cell transduction domains (CTDs) [19]. Some diseases and biological processes require biochemical control from within the cell. CPPs can reach specific cellular sites and can facilitate the delivery of associated molecules for therapeutic or imaging purposes [20,21]. CPPs are able to load and deliver different molecules, such as peptides, DNAs, siRNA, and drugs, so they can simply function as delivery vectors [19,20,21,22,23]. The anchoring of the target molecule to the CPP can be done either via covalent or noncovalent electrostatic or hydrophobic interactions [24]. In some cases, the latter is preferred, especially when the covalent attachment might jeopardize the biological activity of the payload [24].

TAT is a regulatory protein expressed from the HIV long terminal repeat (LTR) and it is essential for viral replication. It is composed of 86 amino acid residues comprising two Lys, six Arg in addition, to seven Cys over sixteen residues (Figure 2) [25]. Frankel and Pabo noticed that the TAT protein from the human immunodeficiency virus-1 (HIV-1) could be taken by the cells grown in tissue culture and hence transactivate the viral promoter [25].

TAT (GRKKRRQRRRPPQ) is a cell-penetrating peptide identified in a protein component of the HIV-1 virus. The peptide is widely used as a vector for the transport of active molecules into cells and nuclei [27,28]. A shorter TAT peptide sequence has also been shown to have cell-penetrating properties in several studies on cell internalization and gene delivery [29,30]

Green and Loewenstein have synthesized TAT, and a mutant version of it, and demonstrated the fast uptake of the TAT by cells [26]. Interestingly the mutant versions (21 to 41) of the amino acids also showed significant activity (Figure 2). The author suggested that two regions are operational: one for the transactivation and the other for the binding or nuclear targeting region. It is to be mentioned that the aim of all these studies is to study how to disrupt the functionality of the TAT protein to stop the viral replication process that TAT is considered a main part of [26].

Various studies have shown the ability of TAT to translocate through the plasma membrane and then transactivate the viral gene. A fragment of the TAT-derived peptide (37 to 72) was proven to be able to accomplish internalization into various cells and tissues. This fragment comprises basic amino acids which are believed to be responsible for the translocating activity. Furthermore, the fragment exhibits an α-helical structure with amphipathic characteristics. This property could be a key feature for the uptake of this peptide by the enveloped viruses either via fusion or endocytosis mechanisms [26].

Vivès et al. have [31] synthesized several peptide fragments from the shortest sequence that overlaps with two major domains, which are believed to be involved in the cell translocation and nuclear targeting (37–60) of the TAT peptide (Figure 2). The aim of their study was to establish the main domain responsible for translocation and nuclear localization capabilities. The selected sequence has two consecutive Pro residues at its *C*-terminal located between highly basic residues and the Cys residue. This sequence could act as a spacer between the peptide and the payload, leading to a favorable interaction between the peptide and the cells, resulting in the desired translocation and uptake. Peptide fragments were synthesized, including the selected sequence 37–60, with deletions made at *C*- and *N*-termini, as follows: the *N*-terminal part 38–49 is considered to possess an amphipathic α-helical structure; thus, two fragments were synthesized, 43–60 and 48–60. The *C*-terminal is considered to be rich with basic amino acids, thus the 37–53 fragment was also synthesized. Interestingly, the 43–60 and 48–60 peptide fragments retained their cell internalization and nuclear accumulation, suggesting the crucial role of the basic domain, as they contain the full basic domain and deletions in the helical domain. On the other hand, the 37–53 fragment was not taken in by the cell even at concentration as high as 20 µM [31].

Surprisingly, the fragments with an α-helical characteristic (37 to 53) seemed not to be crucial, neither for cellular uptake nor the nuclear translocation. On the other hand, all the fragments with basic residues were taken up by cells in less than 5 min with a concentration as low as 100 nM. Furthermore, the study revealed that the helical structure could adversely affect the internalization induced by the basic domain; it slows down the diffusion of the peptide into the cell and towards the nucleus. It is noted that this peptide lacked three Arg residues and Pro-Pro-Gln at the *C*-terminal, so the authors synthesized another analog to investigate if this defect in the translocation or nuclear targeting is ascribed to the absence of those residues or to the helical structure. The tests confirmed that the absence of those residues had no effect on the reduced efficiency, as no enhancement in the cell uptake or translocation was observed in comparison with the 48–60 fragment. The authors proved that the free sulfhydryl of the Cys residue had no effect on the translocation or localization by comparing the free and the protected sulfhydryl group, and no difference in the functionality was noticed [31].

Wender and coworkers demonstrated the significant role of the (TAT 49–57; RKKRRQRRR) fragment in the translocation process [32]. Further modifications to this sequence, including Ala substitution, resulted in a nonfunctional peptide in terms of cellular uptake. A 9-mer of Arg showed a 20-fold higher cellular uptake with respect to the TAT 49–57, suggesting that the main factor is not the charge itself, but rather the chemical characteristics of the residue itself. Moreover, the 9-mer of the D-Arg showed even a higher uptake rate (100-fold). Obviously, the effect of the guanidium motif is responsible for such effect. This research group designed various polyguanidine-based analogs which revealed an enhanced uptake rate and protease resistance in comparison to either TAT 49–57 or the 9-mer Arg [32].

Park and coworkers confirmed that the Arg and Lys rich sequence (TAT 49–57; RKKRRQRRR) is responsible for the protein transduction through the plasma membrane [33]. Their study showed that any additional deletions at either the *C*- or *N*-termini caused a reduction in the transduction capability. The authors fused green fluorescent protein to the previous peptide (TAT 49–57), in addition to nine consecutive Lys residues once and to nine consecutive Arg residues. The rationale behind this selection is the presence of several positively charged residues with the main sequence (six Arg, two Lys, and one Gln), which are believed to have a crucial role in the transduction activity [33].

Interestingly, polylysine (9Lys-GFP) and polyarginine (9Arg-GFP) showed a comparable transduction effect as well as nucleus localization with the main aforementioned sequence. These proteins were successfully delivered to the cell with the same transduction capacity. The importance of the positive charges in the peptide sequence can be ascribed to the interaction between those positive charges and the negative ones on the surface of the cell membrane. In conclusion, this study reaffirms the importance of the basic domain of the TAT peptide for the transduction activity [33].

It is noteworthy that the transduction is also dependent on the characteristic of the fusion target protein. Wender et al. reported the higher transduction efficiency of the 9-mer Arg peptide with respect to TAT 49–57 [32]. This difference is mainly ascribed to the difference in the fusion proteins in both studies. It is to be noted that, in the previous study, the Gly residue at position 48 of the TAT peptide was believed to be part of the nuclear localization signal (NLS). However, as the polyarginine, polylysine, and TAT 49–57 peptides could transduce the GFP and translocate into both the cytoplasm and the nucleus, similar to TAT 48–57 peptide, this clearly refutes the Gly role with respect to NLS [33]. Some studies have stated that the denatured protein containing TAT transduces more efficiently than the correctly folded ones; this behavior could be ascribed to the reduced structural constraints, which enhances the passage of the protein through the cell membrane [34,35,36].

Various mutational analyses of the TAT protein to study its activation mechanism of the human HIV-1 made it possible to propose a functional organization of this protein [25]. In short, the cysteine-rich region of TAT (between 21–38) is important for metal binding, and any sequence mutations in the basic-rich regions would obstruct its functionality [25]. However, studies showed that the metal binding is not required for TAT at the transcription stage [37]. Nevertheless, whilst the activation mechanism of the TAT protein is not fully understood, the binding to the target nucleotide is an important prerequisite for the transactivation to happen [26]. Furthermore, TAT usually accumulates in the nucleus [38]; however, mutating the base rich region of TAT protein (48–52) resulted in a nonfunctional TAT in the cytoplasm [26].

Surprisingly, when other research groups tried to repeat previous studies using shorter versions of the TAT protein, it seems that these selected shorter versions were unable to replicate the performance with the previous group who originated these mutations. Some groups have highlighted the importance of having some periodicity in the acidic, polar, and hydrophobic residues for the TAT to be functional. Loret and coworkers have examined this point by selecting two sequences (2–23 and 38–60) and tried to induce the helicity in them [39].

These sequences were selected as they contain two candidates of activating regions. Circular dichroism (CD) experiments showed that only the 38–60 peptide fragment adopts the α-helical structure in 90% 2,2,2-trifluoroethanol (TFE), whereas in the aqueous solvent, a random-coil structure was obtained. On the other hand, the 2–23 peptide fragments adopted the random-coil structure in both solvents. Molecular dynamic calculations showed that higher temperature caused stretching of the cationic *C*-terminal of the 38–60 fragment out of the helical structure. The study proved that the 1–14 fragment could be an activating region, not due to the helicity factor, but rather as a Pro-rich region [39].

Peptide transduction domains (PTDs) can facilitate the delivery of potential compounds with promising biological activity in vitro. However, they fail in vivo due to physicochemical constraints, including their size and their lipid solubility. The PTD approach is a smart method which can deliver promising molecules with low or even no intrinsic bioavailability [40].

Comparing the specific PTDs with short functioning sequences suggests that having positively charged amino acids (Arg and Lys) are important for cell permeation and/or interaction with lipids. Those sequences are not structured; they have random coiled structures. However, protein structure prediction algorithms suggests that TAT can adopt helical structures, and the ANTP has a helical structure when present in the homeodomain. As peptides can be transduced even at 4 °C, this means that the absorptive endocytosis is not the mechanism and no receptor is needed. Therefore, any cell can potentially be targeted [35]. Interestingly, when the cells were treated with a drug that inhibits the cellular uptake, such as brefeldin A, the transduction was not affected. Thus, the precise mechanism of the transduction remains a difficult question to answer [35].

The Dowdy group have considered an 11 amino acid fragment of the original TAT peptide (YGRKKRRQRRR) [36]. The transduction occurred rapidly in a concentration-dependent manner and independent of receptors and transporters, and instead is thought to target the lipid bilayer of the cell membrane [36]. The authors investigated the in vivo performance of this fragment by attaching a tetra Gly peptide and a fluorophore (fluorescein isothiocyanate, FITC) FITC-GGGG. It was observed that this motif was quickly transduced into almost all the cultured cells. Next, the FITC-GGGG-11-amino acid TAT peptide was administered into mice along with a control sample of the FITC only. After 20 min, a high-fluorescence signal was obtained in all the blood cells. However, the blood cells of the mice administered with the control sample of FITC showed a slight constant increase in the fluorescence intensity. The TAT-FITC peptide was also transduced in all the splenic cells, the brain cells, as well as skeletal muscle, whereas the control sample remained in the background levels only [36]. This group investigated the possibility of delivering larger biological molecules by fusing a 116 KDa β-galactosidase to the 11 amino acid TAT peptide fragment, resulting in a 120 KDa β-galactosidase-TAT PTD. The control fragment was missing the PTD (11 amino acids part). Both were labeled using FITC. FITC-labeled β-galactosidase-TAT PTD was detected inside the cells and transduced to the maximum intracellular concentrations within less than 15 min, whereas the FITC-labeled β-galactosidase alone was not detected even after 2 h. Injecting these molecules into mice demonstrated the presence of the FITC-labeled β-galactosidase-TAT PTD in all blood and splenic cells, while the FITC-labeled β-galactosidase was not detected. Further successful experiments with other payloads were demonstrated; TAT-Cdk2-DN (human Cdk2 (cyclin-dependent kinase-2) dominant negative) (36 kDa) and TAT-CAK1 (yeast CAK1 (Cdk-activating kinase-1)) (47 kDa) [36].

Joliot et al. synthesized a 60-amino-acid sequence that corresponds to a Drosophila antennapedia complex called homeobox (pAntp) [41]. It is capable of translocating across the neuronal membranes and reaching the nucleus. Isothiocyanate was attached to pAntp to enable the tracing of the pAntp and its penetration ability. A strong fluorescence signal was observed, which suggests that the peptide may enter the cells by nonspecific diffusion or by any uptake mechanism. As pAntp is a basic peptide (pKa 11.35), it could bind to the cell surface and enter only during the course of fixation. The authors investigated this possibility by examining the fluorescent pattern in nonfixed living cells. The results confirmed the internalization of pAntp by live cells. Moreover, after treating the cells with proteinase K, which completely degrades the cell surface molecule neural cell adhesion molecule (NCAM), the fluorescent pAntp was still seen in the nucleus. This confirms that the nuclear localization is not attributed to fixation [41].

Derossit and coworkers investigated several shorter sequences of the antennapedia (60-amino-acid lengths) [42]. In their previous work, they generated a mutant of pAntp called pAntp48S by replacing the Gln at position 50 within the third helix to Ser and deleting two hydrophobic residues (Phe^48^ and Trp^49^). In contrast to the wild-type homeodomain, pAntp48S is not internalized by the nerve cells. The authors concluded that the *C*-terminal region, and specifically that of the third helix, is key for translocation through the plasma membranes. They synthesized various analogs of the C-terminus of pAntp [42]. The authors noticed that the 16-amino-acid sequence derived from the third helix of the homeodomain (RQIKIWFQNRRMKWKK) called pAntp (43–58) as well as the 20 amino acid one are both able to translocate through the membrane at 4 °C, just as with the parent 60-amino-acid one by an energy-independent mechanism. On the other hand, sequences of 15 amino acids are not internalized by cells (Figure 3) [42].

The behavior of these peptides was evaluated on 12–22% sodium dodecyl sulfate (SDS)-polyacrylamide. The peptides showed a tendency to aggregate in the presence of SDS and were eluted as dimers. No aggregates were observed in an acidic gel with the absence of SDS, and they were eluted as monomer. The behavior of peptides in a detergent environment could represent that within the lipid bilayer [42].

These peptides at 25 µM were left in contact with the cells for 3.5 h at 37 °C. Peptides 41–60 and 43–58 are internalized where a strong biotin accumulation was observed in the nuclei. Other shorter peptides, 46–60 and 21–55, are not, or are poorly, internalized. To confirm the internalization of 41–60 and 43–58 peptides, freshly dissociated nerve cells were incubated for 2.5 h at 37 °C in a culture medium that contains the peptides at 70 µM. This study confirmed the internalization of both peptides. Moreover, both peptides could be recovered undegraded and with multimeric complexes, reflecting the aggregation enhancement by the presence of lipids. On the contrary, the other two shorter peptides, 46–60 and 41–55, were not present in the cell extracts. Interestingly, the internalization offers a shield against the proteolytic attack, as confirmed by treating the peptides in the cell culture with 0.25% trypsin for 15 min at 37 °C before or after the incubation with cells [42].

Incubating both the 41–60 and 43–58 peptides at low temperature 4 °C showed only little internalization inhibition, whereas the low temperature completely inhibited the little internalization of 46–60 and 41–55 peptides. Lower peptides concentrations were also investigated to understand whether the concentration has an effect on the internalization. Cells were incubated for 2.5 h with various concentrations (from 1.25 to 20 µM). The high concentrations do increase the internalization, whereas the translocation does happen at all concentrations. Peptide 43–58FF, with two Phe residues replaced by two Trp residues at the 48 and 56 positions, showed less internalization efficiency than the original 43–58 peptide analog. Furthermore, 43–58FF showed less tendency to form aggregates, as per the gel analysis [42]. In conclusion, the entire homeodomain crosses the biological membrane more efficiently and its accumulation level is even higher, although it has been administered at lower initial concentration. This behavior could be ascribed to the stronger binding of the entire homeodomain to the target genomic sequence than the shorter peptide versions. The 20- and 16-amino-acid sequences could be recovered intact, suggesting that they were not targeted at the lysosomal compartment. Interestingly, swapping two Trp residues with two Phe diminished the translocating capability. Thus, the internalization is not only attributed to the hydrophobicity only, but it is sequence-dependent as well [42].

The penetratin peptide (RQIKIWFQNRRMKWKK) is another natural CPP. It is derived from a DNA-binding protein in Drosophila fly morphogenesis and has an ability to translocate across neuronal membranes [43]. Liu and coworkers investigated various CPPs for their penetration (RQIKIWFQNRRMKWKK) ability as well as their cellular uptake. The study considers the noninvasive intraocular delivery. Penetratin (derived from a nonviral protein) outranked TAT (GRKKRRQRRRPPQK), 8-mer Arg (R8), low-molecular-weight protamine (VSRRRRRRGGRRRRK), and 8-mer Ser (S8) at both low and high concentrations in the study [44]. The tissue toxicity was carried out by culturing human conjunctival epithelial cells (NHC) with the peptide solution under investigation for 12 h. All the tested peptides showed a low cytotoxic effect at concentrations below 0.3 mM. The lowest tissue toxicity was obtained in the case of penetratin, with an IC_50_ of higher than 2.5 mM. The most cytotoxic CPP were R8 and protamine, with an average IC_50_ of 0.7 mM. TAT had an IC_50_ of 2 mM, which was similar to the control peptide S8, with an IC_50_ of 2.7 mM. Cellular uptake was determined using fluorescence microscope, where 5-carboxyfluorescein (FAM) was conjugated to the CPPs and a 285 µM concentration FAM-labeled CPPs was considered. No damage or change in the morphology of the NHC cells was observed after treating them with the CPP solutions for 4 h (similar to untreated cells) [44].

The CPPs nonspecifically saturated the cells in a time-dependent manner, with outstanding conjunctiva cell uptake levels being observed for penetratin. TAT was absorbed within 30 min, while protamine and R8 were lower than both penetratin and TAT peptides, but higher than the negative control S8 peptide. Penetratin showed 28- to 153-fold higher cellular uptake than the untreated group and 16- to 29-fold higher cellular uptake than 8-mer Ser (neutral pseudopeptide). The S8 group showed higher (but modest) uptake than the untreated group, and while the group that was treated with the S8 peptide showed a weak fluorescence signal, the groups treated with CPPs showed a much stronger signal, which was localized in the epithelium and sparsely throughout the corneal stroma. The fluorescent pattern was nonspecific and consistent with the cellular uptake data (penetratin > TAT > protamine > R8). After only 10 min, the penetratin reached both the anterior and posterior segments of the eyeballs. The signal reached its peak at 30 min, then started to decrease, but the fluorescent signal was still observed after 6 h. In contrast, only a weak signal was observed in the case of the S8-treated eyes, and the signal diminished within 2 h. CD data showed the S8 had a random-coil conformation, as was the case of protamine and R8, and also for TAT, but to a lesser extent, while for penetratin, its conformation is a type II (PPII) helical structure. The role of the positive charge of the peptide was confirmed, as the neutral control 8-mer Ser pseudopeptide showed weaker permeability with respect to the rest of the examined peptides. However, positive charge was not the only factor that governs permeability. The most efficient peptide in this study, penetratin, was not the most positively charged one. It is worth mentioning that, owing to the amphipathic character of penetratin, it was able to utilize its hydrophilicity while interacting with the continuous secreted tear film, which naturally prevents molecules from getting access to the eye. On the other hand, its lipophilic part was considered to interact with the corneal epithelium. Furthermore, its corneal permeability was 87.5 times more, hitting the threshold of small molecule drugs [44].

In a prestigious study carried out by Patel et al., who compared the capacity for delivering green fluorescent protein (GFP) via different CPPs and their cyclic analogs [45], the following peptides were considered: penetratin, R8, TAT, transportan, and xentry (LCLRPVG). The GFP was selected as a cargo due to its fluorescence property, which can be easily traced and quantified. The cellular uptake was monitored in four different cell lines of different tissue origins (HeLa—human cervical epithelial, HEK—human kidney epithelial, 10 T1/2—mouse embryonic, HepG2—human liver epithelial) [45]. The study revealed that the cellular uptake was dependent on the sequence of the CPP and whether it is linear or cyclized, as well as on the cell line. Interestingly, the special subcellular profiles of the GFP were also different among the different CPPs. HeLa cells were incubated with 10 µM of the CPPs at 37 °C for 1 h. While no CPP was incubated, no fluorescence signal was detected with the confocal microscope. The eGFP-R8 was enriched in perinuclear localization, while the eGFP-transportan was localized towards the periphery of the cells. The eGFP-TAT was similar to that of the eGFP-R8, but with less internalization. As per the flow cytometry analysis, the highest cell uptake of the cationic CPPS (penetratin, R8, and TAT) was for the 10 T1/2 cells. Increased uptake was observed for all CPPS with respect to the internalization by HeLa and HEK cells. The eGFP-transportan was the only peptide to show uptake by the HepG2 cells. The overall performance of CPPs in the different cell lines were as follows: cTAT > cR8 > HA-TAT≈Transportan > R8 > Penetratin ≈ TAT > xentry and 10 T1/2 > HepG2 > HeLa > HEK for the CPPs and for the cell lines tested, respectively. It is worth mentioning that none of the CPPs used caused toxicity in the Hela cells [45]. To investigate the effect of cyclization on the cellular uptake, Cys residue was introduced at both termini in all CPPs (except xentry, which has an endogenous Cys) in order to have the cyclized version of the CPPs form via disulfide bridges. Additional modification was considered to check its effect on the cellular uptake and cytosolic delivery. An endosomal escape sequence (from the influenza virus, HA) was added between the GFP and the TAT peptide. Cyclization and the HA incorporation enhanced the cellular uptake, with the fluorescence signal levels higher in cyclic peptides compared to their linear analogs. Interestingly, the localization was also different in the case of the cyclized GFP-R8 and GFP-TAT with respect to their linear analogs. The cyclization resulted in a more peripheral disposition within the cell. HeLa cells showed the largest uptake in response to cyclized CPP of 4- and 5.9-fold for R8 and TAT, respectively. Furthermore, the addition of HA to the TAT has increased its uptake by all four cell lines. The punctate signal observed in all CPP suggests that endocytosis is involved in the uptake mechanism [45].

The highly cationic nature of the TAT and penetratin peptides is a key to their cell-penetration ability. Indeed, mammalian cell membranes are rich in zwitterionic lipids such as phosphocholines that become polarized in the presence of cationic peptides. This effect is more pronounced with arginine-rich peptides, as they contain guanidinium groups, forming bidentate bonds with phosphate moieties in the lipid molecular structure [46]. Upon interaction with the guanidinium group, arginine-rich cell-penetrating peptides cause the formation of negative Gaussian curvature in the lipid membrane, thus facilitating membrane pore formation [47,48]. Besides natural cell-penetrating peptides, synthetic oligoarginines, such as octa-arginine and cyclic arginine-rich peptides, have been investigated as vehicles for cellular transport [49,50].

Galparan (Transportan) is a chimeric peptide composed of 27 amino acids. It comprises 13 amino acids from the *N*-terminus of the galanin neuropeptide and 14 amino acids from the *C*-terminus of the wasp venom peptide toxin (mastoparan) [51]. Mastoparan penetrates the cell membrane and translocates into the inner leaflet, creating short-living pores. Galanin shows various inhibitory effects such as inhibiting the glucose-induced release [51]. Pooga and coworkers have studied the cellular uptake and the localization on the cell surface of the galparan. They synthesized the [Lys^13^] analog of galparan [52]. Due to high cellular uptake of this peptide, and the available active side chain of the Lys amino acid, which could be used to anchor various cargos to this peptide, the authors coined the name of this novel peptide as Transportan (Figure 4).

N^ε13^-biotinyl-Transportan was shown to be internalized following an energy-independent pathway. The internalization took place efficiently at (37, 4, and 0) °C. In addition, the cellular uptake could not be blocked by treating the cell with various cellular uptake inhibitors [52]. Biotinyl–Transportan was detected throughout the Bowes’s melanoma cells’ interior. Transportan was accumulated in the plasma and nuclear membranes. The peptide was initially internalized quickly at 37 °C. after first 5 min, and it was localized in the plasma membrane and the cytosolic membranous structures (endosomes, endoplasmatic reticulum, and Golgi). There was also some localization in the nucleus, albeit the stain was bright but visible. After 15–30 min, the biotinyl–Transportan was preferentially localized in the nuclear membrane and nuclei. The peptide was also detected at 0 and 4 °C in all membranous structures, but mainly in the plasma membrane. Increasing the incubation time for 1–2 h made the peptide detectable in the nuclear membrane. Thus, at 0 and 4 °C, either the peptide was not penetrating or its concentration was below the detection limit. The penetration feature was noticed in different cell lines as well; however, the authors did not disclose this data [52].

Treating Bowe’s melanoma cells with either hyperosmolar solution of sucrose (which blocks the formation of clathrin-coated pits) or phenylarsine oxide (which crosslinks the thiol groups of membrane surface proteins) did not affect the internalization of biotinyl-transportan. Thus, this confirms that the transportan is not internalized via receptor-mediated endocytosis pathway. It is worth mentioning that after treating Bowe’s melanoma cells with phenylarsine oxide, some cells detached from the surface, however, the remaining cells took the biotinyl–Transportan efficiently. Interestingly, galanin internalization was lost in response to treatment with phenylarsine oxide (the author did not publish this data). The study showed that the internalized biotinyl–Transportan is protected against degradation by trypsin. On the other hand, incubating it with trypsin under the same conditions abolished any uptake, confirming that the trypsin completely digested the biotinyl–Transportan. [52] The ^125^I-biotinyl-transportan concentration in water was 15-fold higher than in octanol, and independent of concentration. This finding does not contradict its affinity to the membrane, providing that the partition coefficient of ^25^I-biotinyl-transportan is six-fold more than the ^25^I-ion from ^25^INa. Caution must be taken as the octanol/water model does not accurately represent the interaction between the peptide/phospholipids of the membrane. The authors suggested that the internalization of biotinyl–Transportan follows that of antennapedia (inverted micelle mechanism) [42], in which the presence of at least one Trp residue is important. The biotinyl–Transportan has Tyr and Trp residues in its chain, which are important for the micelle formation purposes [52].

A study carried out by Hirose et al. postulated that the peptide penetration is accompanied by the formation of “particle-like” multivesicular structure on the plasma membrane, along with a topical inversion of the plasma membrane [53]. The study used Alexa Fluor 488 conjugated to a 12-mer Arg peptide (R12-Alexa488). Treating cells with the unlabeled peptide, very few species were observed within the plasma membrane, even after using high concentrations. This confirms the role of the Alexa 488 fluorophore as a hydrophobic moiety. Other hydrophobic moieties were also attached to the peptide, confirming these findings. The study showed that attaching R12 peptide to hydrophobic moieties (fluorescent moiety or a peptide tag derived from the human influenza hemagglutinin (HAtag YPYDVPDYA)) helps in stimulating certain dynamic morphological alterations in the plasma membrane that allow the peptide to permeate through the plasma membrane [53].

This study was based on the hypothesis that the peptide diffusion happens at specific sites of the plasma membrane. The penetration of the peptide into the cell happened through specific sites, as confirmed by strong florescent signals, and then spread to the rest of the plasma membrane. Differential interference contrast microscopy revealed that the internalization of the peptide was accompanied by the formation of a particle-like structure. The locations of these particles were matching with the sites of peptide infusion. As the R4-Alexa488 peptide had poor translocation ability, no significant alterations in the plasma membrane were observed. The study confirmed the occurrence of the translocation even at 4 and 15 °C, which means that the translocation is an energy-independent process, provided that the addition of macropinocytosis inhibitors did not affect the particle-like formation [53]. The authors studied the effect of the membrane potential on the particle formation. The high Na^+^ concentration is considered a normal condition that maintains the membrane potential (physiological condition). On the other hand, having K^+^ in high concentrations instead diminishes the membrane potential. Using an Na^+^-rich buffer to incubate the cells showed the expected particle-like formation, whereas this was not the case when having the K^+^ as predominant in the extracellular medium, even when high concentrations of the R12-Alexa488 were considered. Interestingly, treating the cells with an excess of Na^+^ after they were treated with an excess of K^+^ showed a recovery in the influx of the peptide [53].

In summary, the interaction of the peptide with the plasma membrane is also dependent on the membrane potential and not only the peptide itself. Membrane inversion was also accompanied in the influx process. Phosphatidylserine is a membrane component that is localized to the inner side of the plasma membrane at the normal conditions. On the other hand, under certain conditions, it could be interrupted, such as in the case of apoptosis, in which it would be inverted, which could be investigated using specific binding partner. Such membrane movement (inversion) was observed with the R12-Alexa488 and not with the R4-Alexa488 one [53].

The study highlighted some dynamic rearrangements of the membrane with the interacting molecules. For example, the acidic constituents interacted with the basic residues of the R12. Scanning electron microscope (SEM) analysis of the plasma membrane upon penetration of R12-Alexa488 revealed the presence of various small vesicles instead of a single large one. However, eventually the accumulation of the peptide will lead to the formation of large particles at the site of influx. Such observation was not seen in the control cells (untreated cells). Transmission electron microscope (TEM) analysis confirmed the formation of the particles at both low and physiological temperature (4 and 37) °C, respectively. This work confirms that the structural alterations in the plasma membrane are endocytosis-independent and happened as a result of the physicochemical interactions between the peptide and the plasma membrane [53]. Interestingly, as the study revealed no leakage of the lactate dehydrogenase from the treated cells, this means that the plasma membrane retained its integrity. Several membrane-repair mechanisms are operating at such conditions. However, the authors could not determine which was operational here. However, they confirmed that neither lysosome nor endosome-mediated membrane-repair was taking place. R4-Alexa488 has low permeability and thus no significant alterations in the plasma membrane. On the other hand, treating HeLa cells with this peptide in the presence of unlabeled R4, R12, and R12-HAtag led to a noticeable increase in the cellular uptake of R4-Alexa488, in addition to membrane-particle formation. Treating the cells with R4-Alexa488 peptide with unlabeled R12 showed only little increase in the cellular uptake. The addition of unlabeled R4 did not increase the cellular uptake of R4-Alexa488 [53]. The authors checked whether the particle formation was due to the simple interactions between the peptide with the lipids or not. They studied the R12-Alexa488 with giant vesicles that mimic the composition of the plasma membrane. No peptide translocation was observed, confirming that the particles formed are not due to a simple interaction between peptides with lipids [53].

El-Andaloussi and coworkers have studied the delivery capability of M918. M918 is a 22 AAs peptide (MVTVLFRRLRIRRACGPPRVRV-NH_2_) derived from the tumor-suppressor protein p14ARF [54]. It comprises the fragments from 1–22, with inverted positions for 3–8, with seven positive charges. Initially the peptide was prepared as a control to mimic the activity of the ARF protein. Surprisingly, it showed an excellent penetration capability [54]. These authors investigated the following five CPPs versus the M918: penetratin, transportan 10 (TP10, AGYLLGK*INLKALAALAKKIL-NH_2_) a 21 AAs that represents a deletion analog of Transportan, YDEGE, M705 PNA (CysKKCCTCTTACCTCAGTTACAKK^+^-NH_2_), and M705 inverted PNA (CysKKCCTCTTACACTCGTTACAKK^+^-NH_2_) [54]. The results showed the better internalization of M918 than penetratin and TP10 f HeLa cells and Chinese hamster ovarian (CHO) cells. YDEGE was used as a negative control (it showed negligible uptake) and to roll out any uptake effect as a result of membrane disruption. Moreover, no lactate dehydrogenase (LDH) leakage was observed, except in the case of TP10, where a massive membrane leakage was observed at 25 µM. M918 did not show any cell proliferation after 72 h with any of the cell lines under investigation. As the high cellular uptake could be ascribed to the peptide aggregation in the cellular membrane, the authors analyzed the uptake using the confocal microscope. No clusters of the peptide were observed in breast cancer MCF-7 cells or HeLa cells. Some membrane-associated peptides were observed in astrocytoma cells, Hifko. The majority of the internalized materials reside in the vesicles, and a negligible amount in the cell nucleus. M918 was further studied and checked for its delivery potentials [54]. Avidin and streptavidin were chosen as model proteins for the transduction experiments. Physical mixtures of streptavidin or avidin with penetratin, TP10, and M918 peptides (noncovalent complex) proved that the three tested peptides were able to promote the cellular uptake of fluorescein-labeled streptavidin at a 5 µM concentration, with the M918 being superior to penetratin and TP10 [54]. The same peptides were tested at a 1 µM concentration; however, with no enhancement in the delivery. On the other hand, with 1 µM of the biotinylated peptides, a noticeable uptake enhancement was observed, in particular with TP10 and M918. Penetratin was low in both strategies. The study proved the superiority of M918 to deliver antisense PNA-targeting pre-mRNA over TP10 and penetratin. Confocal microscope study confirmed that the distribution of the peptide is within the cells and no clusters were observed on the cellular membranes. Therefore, the high cellular uptake is attributed to the internalized peptide rather than to the aggregation in the cellular membranes [54]. Cells were treated and incubated with M918 at 4 °C. The cellular uptake was inhibited at low temperature (4 °C) in the HeLa cells. The same was noticed in the CHO cells when treated with penetratin and TP10. The addition of endocytosis inhibitors, cytochalasin D, which blocks the actin polymerization that is important for the formation of micropinosomes and wortmannin, which is a PI3-kinase inhibitor that also inhibits clathrin-mediated endocytosis, resulted in a decreased uptake of M918. Thus, the authors concluded that the endocytosis pathway (macropinocytosis) mediated the uptake, at least partially. Some studies emphasized the necessity of glycosaminoglycans (GAGs) on the cell surface for the uptake process. While using the GAG-deficient CHO cell, a significant decrease in the uptake of TP10 and penetratin was observed with respect to the wild-type CHO cells. On the other hand, only a little decrease in the uptake of M918 was observed when the cells were treated with the three peptides which were covalently linked to fluoresceinyl-labeled PNA (via a disulfide bridge). The luciferase was measured after 16 h considering 5 µM concentrations of peptides [54].

M918 proved to be the most potent vector, where a 15-fold increase in the splicing was observed versus 10- and 5-fold for TP10 and penetratin, respectively, with respect to the untreated cells. Most importantly, neither inverted PNA conjugated to M918 nor naked PNA induced significant splicing at 10 µM, which proves the importance of the CPPs for cellular uptake. As the endosomal entrapment is a major concern for the endocytosis pathways, the authors were able to circumvent this concern by adding a lysosomotrophic agent (chloroquine), and the splicing was improved with respect to M918-PNA without the chloroquine [54].

## 3. Stapled Peptides

Several chemical strategies could be used to enforce α-helical structures, including hydrocarbon stapling. The α-helical structure plays an important role in preserving the stability of peptides against proteases [55]. Hence, preserving this conformational property is of utmost important to enhance the half-life of the peptide and its binding affinity as well. Moreover, as a result of preserving helicity, the hydrocarbon stapling strategy slows the degradation kinetics by preventing the cleavage within the stapling site. Their efficacy has been demonstrated by various studies, which are discussed here.

Several crosslinking strategies are available for α-helix stabilization [56], such as disulfide bridge formation [57] and lactams formation [58]. The study by Verdine and coworkers was derived from the chemistry of olefinic crosslinking of helices reported by Grubbs and coworkers [59]. In this approach, a ruthenium-catalyzed ring closing metathesis (RCM) reaction is performed to crosslink helices via *O*-allyl serine residues located on adjacent helical turns (Figure 5A) [59].

However, Verdine’s group introduced an alternative metathesis crosslinking approach (Figure 5B) because the main approach of Grubbs showed no improvement in the helical stability [60]. They designed unnatural amino acids with either *R* or *S* stereochemistry of the α-carbon and various lengths of alkyl groups. These synthetic amino acids were then incorporated at one or two turns (i and i + 4 or i and i + 7). A methyl group was included in these amino acids to avoid any helix destabilization of the D-amino acids.

Liu and coworkers modified the ascaphin-8 peptide through the stapling strategy. They synthesized several stapled analogs at different positions [61]. The authors observed that stapling at the i + 7 positions delivers more helicity than what could be achieved in the i + 4 analogs, which is ascribed to the longer hydrocarbon chain involved (Figure 6).

All their new analogs showed higher stability as well as higher cytotoxicity against the two cancer cells with respect to the parent linear peptide with no evidence of hemolytic activity (HC50). The synthesized peptides showed the ability to kill the A549 and prostate cancer cell line [61]. The peptides proved to be good antiviral defenders. They were able to disrupt and inhibit the interaction between the receptors that are responsible for providing clearance routes to the host cells (protein–protein interactions). They play a pronounced role in inhibiting the viral activity of the hepatitis C virus (HCV) by blocking the interaction between the HCV envelope glycoprotein E2 and the CD81 receptor, which is in charge of the virus gaining access into the cell [61].

Dhanasekaran et al. identified and synthesized two linear peptides to disrupt the interaction between the CD81 receptor and HCV E2 (PSGSNIISNLFKED and GSSTLTALTTSVLKNNL) [62]. These peptides are basically fragments (176–189) and (158–174) which form the extracellular loop of CD81, respectively. The first peptide (PSGSNIISNLFKED) showed an ability to inhibit the interaction between rHCV-E2 and hCD81 expressed on Molt-4 T cells, but at quite high concentration of 3.5 mM, which is ascribed to the random-coil conformation adopted in water. The authors were able to achieve α-helical structure of this peptide with the aid of trifluoroethanol (TFE), which is known to enhance the formation of the secondary structure. The complete α-helical structure stability was achieved at 30% TFE [62].

Cui et al. modified the peptide that was introduced by Dhanasekaran and coworkers by means of hydrocarbon stapling strategy [63]. The sites selected sites for the stapling were able to leave the active amino acid residues that interact with HCV E2 intact (Figure 7).

CD studies showed that the linear peptide does not have helicity, which confirms the poor efficacy of the linear version that was adopted in Dhanasekaran’s work [62]. The study showed the i and i + 7 stapling positions deliver more helicity than the i and i + 4 positions, and hence would stabilize the α-helical structure. It is worth mentioning that these findings are in line with the work of Liu and coworkers [61]. Furthermore, the stapled analogs showed significantly enhanced half-life with respect to the linear one; 200 min versus 10 min, respectively. All of the stapled analogs showed better inhibition of the HCV than the linear analog. It is to be re-emphasized that the amino acid composition plays a crucial role in the peptide activity. It was expected that all peptides with higher activity would have higher helicity; however, in this study, one of the analogs showed lesser activity despite its higher helicity. This variation is ascribed to the fact that the amino acid of this analog was replaced with an unnatural amino acid (Fmoc-S_5_-OH), and it seems that the replaced amino acid was a key residue to interact with HCV E2 and it should have not been swapped with the stapling hydrocarbon residue. A cytotoxicity study was carried out for the highly promising analog with respect to its inhibitory effect and for the linear analog. Both analogs proved to be nontoxic at the inhibitory concentration of 1000 µg/mL.

Striking protease stability was demonstrated by Bird et al. [64]. They employed a double-stapling approach to a HIV-1 fusion inhibitor. In addition to the improved pharmacokinetics, including the oral absorption, the stapling strategy provides a proteolytic shield by enhancing the α-helical structure, which slows the kinetic of proteolysis, and by protecting the cleavage of the peptide within the vicinity of the staple. Enfuvirtide is the first fusion inhibitor that prevents the entry of HIV-1 to the human. It can disrupt the fusion apparatus of the viral by mimicking the heptad repeat 2 (HR2) oligomerization domain of the g41 envelope glycoprotein (Figure 8) [64].

Despite the great safety profile of peptide-based drugs and their high specificity, there are various obstacles which need to be circumvented to help peptide therapeutics to move forward. Specifically, the fast clearance of most of the therapeutic peptides is one of the major drawbacks in their therapeutical utilization. This challenge has been observed in enfuvirtide as well. Swapping proteinogenic amino acids with the nonproteinogenic ones does not always provide the required resistance against the proteolytic dilemma [64].

Stapling is considered as a strategy of choice in several cases, and sometimes double stapling could be required. In addition, there would be no need to replace the natural amino acids with unnatural ones. In this study, the enfuvirtide peptide and its functionally optimized α-peptide derivatives were subjected to chymotrypsin digestion, and the data obtained from LCMS highlighted their fast proteolysis, with half-lives from 2 to 16 min. To enhance their resistance against proteolysis, a stapling strategy at the *N-* and *C*-termini (i, i + 4) positions was considered. Moreover, due to the large chain of the selected peptide, a double-stapled strategy was also investigated, in which two pairs of (i, i + 4) insertions were made (Figure 8) [64]. The modified analogs were subjected to chymotrypsin digestion and a dramatic enhancement in the half-lives was noticed: 77–116 min for the single-stapled analogs and 335 min in for the doubly stapled examples [64]. To validate the role of the double-stapled strategy, the singly and doubly stapled analogs were also generated for enfuvirtide and exenatide. The same findings were found in both drugs (USFDA-approved drugs for HIV-1 and diabetes, respectively), in which the singly stapled peptides possessed longer half-lives than the unmodified parent sequence. Moreover, the doubly stapled ones had much longer half-lives with higher proteolytic resistance [64]. The explanation for the dramatic enhancement of proteolysis resistance in the case of the double-stapled analogs is that the proteases require the peptide to adopt an extended conformation to be able to hydrolyze the amide bond. On the other hand, enforcing the α-helical structure in peptides would shield the amide bond and lead to an increase in their protease resistance. However, it was confirmed by CD analysis that the superior stability against the proteolytic enzymes was not attributed to the degree of the α-helicity. Though the stapled peptides did show higher helicity than the unmodified counterparts, the doubly stapled T649v analog showed an intermediate α-helicity value which was less than that of the single-stapled analogs. The same observations were noted in enfuvirtide [64].

Comparing the effect of inserting the unnatural amino acids in four positions of the peptide, with and without crosslinking them, revealed that the overall α-helical structure stability was the same. However, with regard to the proteolysis stability, it was noted that the unmetathesized analogs were more susceptible to proteolytic degradation, with two to three and eight-fold shorter half-lives than the corresponding single- and double-stapled analogs, respectively. In conclusion, the covalent crosslinking is in charge of the enhanced resistance against proteolysis rather than the overall α-helicity or the added hydrophobicity. Over various temperatures, the single- and double-stapled analogs showed almost the same behavior, with no tendency to aggregate. Thus, the higher stability of singly or doubly stapled analogs could only be attributed to the covalent crosslinks themselves [64].

LC-MS analysis confirmed the resulting cleavage positions, with chymotrypsin for the unmodified T649v as well as the *C*-terminus of the singly stapled analog at four major positions, but, over different times, with the more prolonged being the singly stapled one. Interestingly, no cleavages were noticed in the vicinity of the stapling. The *N*-terminus singly stapled analog showed two cleavage positions with no cleavages within the crosslinked segment. This confirms that the insertion of the hydrocarbon staples prevented the local proteolytic degradation process. A more delayed degradation kinetic was observed in the doubly stapled analogs in comparison to the singly stapled one, in addition to a complete blockage of the proteolytic cleavage at the installation position of the staples [64].

Interestingly, all of the modified analogs showed an enhanced antiviral activity with respect to the parent enfuvirtide molecule. Provided that the most active analog was the doubly stapled one, it is worth mentioning that having demonstrated antiviral activity proves that no aggregation took place in the doubly stapled analog, otherwise its antiviral activity would have been compromised. Intravenous administration of the doubly stapled and the unmodified peptides into mice showed the higher plasma concentration of the doubly stapled analogs, which was detectable for up to 4 h after the dosing took place. On the other hand, the plasma concentration of the unmodified analog was lower than that of the doubly stapled analog and was not detectable 2 h after the administration [64].

Despite having the same pepsin cleavage sites, the observed resistance of the doubly stapled peptides was 200-fold higher than the unmodified T649v peptide. In addition, it remained 80% intact after 12 h of the exposure. This acidic stability encouraged the authors to investigate the oral absorption. In vivo study showed that the plasma concentration of the doubly stapled analog was detectable after the oral administration, whereas the unmodified T649v was not. Such findings could offer the possibility to administer stapled peptides orally. In addition, the study highlighted the striking protease resistance of the doubly stapled modified peptides, either at neutral or acidic pHs. The study also confirmed that the antiprotease activity is not dependent solely on the extent of the helicity, nonnatural amino acid substitution, or the peptide aggregation. Instead, the main reason behind this phenomenon is the hydrocarbon double staples themselves, which play a crucial role in slowing down the degradation kinetics through the helicity and by preventing the proteolysis at the cleavage sites, especially in the vicinity of the staples [64].

Walensky et al. modified a peptide in charge of interacting with BCL-2 family proteins, which are important for apoptosis regulation through the amphipathic α-helical BH3 peptide [65]. Hydrocarbon stapling was applied to modify the BH3 peptide and enhance its helicity. The modified BH3 peptide was called the stabilized-alpha-helix of the BCL-2 domains (SAHBs) (Figure 9).

Interestingly, it showed high helicity, protease resistance, cell penetration ability, and a high binding affinity towards BCL-2. This peptide can activate the apoptotic pathway to kill leukemia cells and inhibit the growth of human leukemia xenografts. Helicity is important in the protein–protein interactions (PPI) and may either stabilize or to interrupt such interactions. The SAHBs peptides showed an enhanced helicity (by CD) from 35 to 87% versus 16% in the case of the wild-type one. This helix structure can shield the amide bond and hence decrease the chances of its proteolytic degradation. The binding affinity was almost comparable between the modified and the unmodified analogs. Furthermore, the fluorescence polarization binding assay showed a more than six-fold enhancement in the binding of SAHB_A_ with respect to the wild-type peptide. A mutated analog of SAHB_A_ (SAHB_A(G–E)_) showed reduced high binding affinity; thus, it was used as a control [65]. The in vitro studies of SAHB_A_ showed a dose-dependent increase in the cytochrome release, whereas a negligible effect was observed with either the wild-type or the SAHB_A(G–E)_) control. The penetration ability was exemplified by incubating Jurkat T cell leukemia cells with fluorescein tagged peptides, the wild-type (BID BH3), SAHB_A_, and SAHB_A(G–E)_. Tracing the fluorescence with confocal microscope and fluorescence-activated cell sorting analyses showed a fluorescent labeling in the modified analog SAHB_A_ but not with the wild-type. The SAHB_A_ analog also showed an ability to activate apoptosis in Jurkat T cells leukemia cells (5 µM of SAHB_A_) after 20 h. Neither the wild-type nor the mutated control showed an increase in the apoptosis in that dosing range [65].

The p53 is a transcription factor peptide that plays a crucial role in inducing cell arrest and apoptosis, hence protecting the cells from malignant transformation. On the other hand, hDM2 controls its levels by direct interaction and binding, which leads to neutralizing its transactivation activity and causing its degradation via the ubiquitylation–proteasomal pathway. Accordingly, Bernal et al. optimized this peptide by enhancing its helicity through the stapling approach [66]. Attention was paid to not swapping the important residues for the hDM2 engagement. The stabilized α-helix p53 (SAH-p53) was synthesized, which demonstrated that this modification showed enhanced affinity towards hDM2 as well as a cell-penetration ability relative to the unmodified analog (Figure 10) [66].

In this study, the I and i + 7 strategy was considered. The CD experiment proved a clear enhancement in the helicity of the stapled peptide versus the wild form, with 10–59% helicity for the various four synthesized analogs versus 11%, respectively. Moreover, the affinity towards the hDM2 was three orders of magnitude higher in the case of the stapled peptide (SAH-p53-4) versus the wild-type, as determined by a fluorescence polarization binding assay test. Furthermore, this analog showed an enhanced proteolytic stability with respect to the wild-type peptide [66]. The overall charge of the peptide was negative at the physiological pH, hence explaining the poor penetration ability when tested on the Jurkat T cells. Thus, Asp and Glu residues were replaced with Asn and Gln, respectively, resulting in four more analogs (SAH-p53-5-8). Interestingly, enhanced helicity was noticed in the new analogs (from 2 to 8.5-fold), in addition the binding affinity towards hDM2 being retained. The best analog with significant helicity, cell-permeability, and binding affinity with hDM2 was SAH-p53-8. The study showed a promising approach in terms of restoring p53 activity and exposing the cancerous cells to apoptosis. One of the developed analogs (SAH-p53-8) proved its ability to restore the levels of native p53 wild-type. Furthermore, this analog showed a dose-dependent inhibition of cancer cells by reactivating the apoptotic pathway through stabilizing the native p53 peptide [66].

A special class of stapled peptides called stitched peptides was reported by Hilinski and coworkers [67]. They reported a new reaction that renders multiple contiguous staples connected by a spiro link junction. This next generation of stapled peptides showed high thermal stability, in addition to superior proteolytic stability and cell penetration capability in comparison to the single-stapled peptides. However, the authors could not confirm whether the increased penetration was due to the increased helicity of the stitched peptides, their increased hydrocarbon content, or both. Nevertheless, they concluded that a new mechanism of penetration with better performance was observed. The authors investigated the stitched approach on polyarginine oligomer (R12). They compared the parent peptide with its i, i + 7 stapled (R12-Stp), i + 4 + 7 stitched (R12-Stc), and i + 4 + 7 unstitched (R12-Unstc). While R12 showed no helicity, the stapled analog had 23% helicity and the stitched one had 59% helicity at room temperature, while the unstitched peptide had only 15% helicity [67]. Computer simulation studies were performed to investigate the impact of the structural dynamics for these stitched peptides. Whilst the unmodified R12 did not show any secondary structure, both the stapled and stitched analogs exhibited a computed helicity of 21.3% and 56.4%, respectively. Interestingly, the helicity of the stitched analog was throughout the entire peptide sequence, though it was localized on certain residues in the case of the stapled peptides. Solvation analysis of the three analogs confirmed the ability of the stapled and stitched analogs to shield the H- bonds of the backbone from competing with water molecules, hence avoiding the unfolding that could facilitate the access of water to the backbone H-bonds [67]. To investigate their penetration ability, the three analogs were labeled with fluorescein and Hela cells were then treated with a concentration of 5 μM at 37 °C for 1 h. Confocal microscope analysis demonstrated that the stitched peptide preserved the penetration ability of the parent R12 peptide. Moreover, an increase in the cellular uptake (as determined by epifluorescence microscopy analysis) was observed by 21% and 75% for the stapled and the stitched analogs, respectively, provided that a lower concentration of 1 μM and a longer incubation time (8 h) were used. The authors highlighted the fact that the stitched analog had the lowest net positive charge. Hence, the positive charge is not the only factor that drives the penetration process. Given that the stitched peptide is resistant to proteolytic degradation, this resistance will lead to a higher cytosolic accumulation of it; therefore, this could be a reason for the enhanced penetration [67]. Treating these three analogs with trypsin for 5 min at 25 °C revealed the complete degradation of the unmodified R12 analog, whilst 41% of the stapled peptide remained intact. The outstanding stability of the stitched peptide was kept 94% intact. More than 50% of the stitched analog remained intact even after 60 min of trypsin exposure. Furthermore, the half-life of the stitched peptide was also higher than that for the stapled, at 64 min versus 3.1 min. Apparently, the presence of the macrocyclic bridge is responsible for the increased half-life because the proteases are physically blocked from accessing the cleavable residues. In addition, these bridges prevents the peptide from adopting an extended conformation needed for the cleavage to take place [67].

## 4. Liposomes

Gramicidin S (GraS) is an amphiphilic cyclic decapeptide isolated from *Bacillus brevis* with antimicrobial activity against *Staphylococcus aureus* (Figure 11).

However, due to its hemolytic activity on the eukaryotic cells, in addition to its low water solubility, its clinical application is limited. A novel water-soluble formulation of GraS in a liposome bilayer enhanced its bioavailability as well as the affinity of peptides towards the cell membranes, providing an effective mechanism to kill bacteria [68]. To circumvent the solubility obstacle of GraS, Desimone and coworkers developed the formulations that can hold the GraS in dipalmitoylphosphatidylcholine (DPPC) and dimyristoylphosphocholine (DMPC) liposomes (L-GraS). Molecular dynamics simulations were also performed to analyze the encapsulation mechanism [68]. These liposome formulations must be designed carefully, such that the liposomal membrane must adapt the conditions of the therapeutic target. For example, it is known that wound-infected areas have a 2 °C higher temperature; thus, modifying the liposomal composition is important to control the permeability and the release patterns [68]. DPPC and DMPC were mixed with various ratios of cholesterol, hydrated, and then extruded. The liposomes vesicles were obtained used a formulation of DPPC:DMPC:Cholesterol (7:2:1). After obtaining the lipid–GraS film, it was extruded several times at a temperature higher than the melting point to achieve the unilamellar and monodisperse vesicles [68].

The positive charges of some residues in the GraS peptide are responsible for the strong interactions with the negative charges of the lipid membrane, leading to disturbing the lipid bilayer and the subsequent pores and/or defects formation, and then cell death. Interestingly, the GraS maintained its antimicrobial activity with the minimum inhibitory concentration of 0.1 mg/mL versus 0.025 mg/mL for the free GraS. According to the cytotoxic behavior of the liposomal peptide, the best relationship between the antibacterial activity and biocompatibility is to have 0.3 mg/mL of the peptide in the formulation [68].

Faya et al. synthesized and designed nine novel antimicrobial peptides (AMPs) for use against methicillin-resistant Staphylococcus aureus (MRSA). They are cationic, which allows for membrane permeation [69]. The peptides were encapsulated in a liposomal formulation containing oleic acid and vancomycin to achieve a pH-responsive drug delivery system. The strategy of the authors is as follows: at the basic pH, an ion pair will be formed from the peptide with the negative charge of the oleic acid, while at the acidic pH, the oleic acid will be protonated, leading to breaking the ion pair and the releasing of the drug [69]. Data filtering technology (DFT) was used to design the peptides in this study. They applied additional filters, such as number of charges, hydrophobicity, and chain length, to the original 86 peptides. Their final group was nine peptides, in which the frequency of LL, KK, and WW was observed in the selected AMPs. It is worth mentioning that all the selected AMPs have a positive charge ranging from +4 to +7, which is important for their interaction with the negatively charged bacterial membrane. The hydrophobicity ranged from 40 to 60% among the selected analogs, which is important for their partitioning between the lipid bilayer of the bacterial membrane and the aqueous core. Seven analogs out of nine possessed an α-helical structure. The two peptides with no helical structure did not show antimicrobial activity. Helicity was concluded to be important for the interaction with the biological membranes [69]. Based on the previous findings, two analogs were selected for all other studies: AMP-2 (H-EKKRLLKWWR-NH_2_) and APM-3 (H-KWWKLLRKKR- NH_2_), in which they are expected to have significant antimicrobial activity. Minimal inhibitory concentration (MIC) was the lowest in these two AMPs with respect to the rest, with 125 µg/mL for AMP-2 and 62.5 µg/mL for AMP-3 against MRSA. AMP-2 had the same MIC value in the case of *S. aureus*, whereas the AMP-3 was even better 31.25 µg/mL. As the vancomycin acts by compromising the integrity of the membrane, and thus enhances the permeability and uptake, it was used as the control [69].

Enhanced permeability was observed with both AMP-2 and AMP-3. Provided that the molecular dynamic simulation showed the closeness of both AMPs to the PO_4_ beads, they were inserted in the bilayer membrane. As for cell toxicity, both AMPs were tested on mammalian cells, and the viability was in the range from 80 to 85%. Interestingly, both bare AMPs and with the liposome formulation showed hemolysis of less than 1% at a concentration of 0.2 mg/mL, confirming their nontoxicity to the red blood cell. The in vitro release of the vancomycin in AMP-2-Lipo-1 showed a more sustained released pattern at pH 7.4 in comparison to pH 6.0, in which, after 2 h, a 1.25% and 16.64% release was observed, respectively. The same pattern was also observed after 8 h (23% and 67.33, respectively). In contrast, in the AMP-3-Lipo-2, after 2 h, the release was less than 30% at both pHs, which could also be considered as a sustained released pattern. After 8 h, the release reached 49.12% at pH 6, while it was only 17.90% at pH 7.4. Antimicrobial activity was assessed for both new formulations, in which AMP-2-Lip-1 showed an MIC of 1.95 and 3.9 µg/mL at pH 6 and 7.4, respectively [69].

AMP-3-Lip-2 showed an MIC of 3.9 and 7.8 µg/mL at pH 6 and 7.4, respectively. The previous findings could be ascribed to the electrostatic linkage between the negatively charged oleic acid and the positively charged AMPs at pH 7.4. While at pH 6.0, such linkage is being broken, as both species have been protonated. As for the naked vancomycin, the MIC was 15.65 µg/mL at both pHs. The superiority of the new formulation with respect to the naked drug could be attributed to cationic effect of the AMP, and hence the enhanced permeation plus the oleic acid and the drug are being released into the therapeutic target in a sustained manner [69].

Infecting a tissue culture with MSRA for 2 h, then applying naked vancomycin or AMP-2-Lip-1 or AMP-3-Lip-2, proved the intracellular activity of the new formulations. Unsurprisingly, AMP-3-Lip-2 (which had the lowest MIC) showed the lowest colony-forming unit (CFU) counts [69].

## 5. Hydrogels

The benefits of hydrogel as wound dressing have been widely agreed upon, especially due to their high water content, which plays a crucial role in the wound healing process. Depending on the chemical structure of the hydrogels, they could be used either as drug carriers or used directly due to their inherent antimicrobial activity. Here, some recent advances in the field are summarized.

Azevedo and coworkers repurposed an already existing antimicrobial peptide into a more effective and safer formulation utilizing a hydrogel approach [70]. Polymyxin B (PMB) is a classic antibiotic for treating drug-resistant Gram-negative bacteria. The hydrogel assembly was done by adding a PMB solution to a solution of amphiphile peptide (PA). Another antibiotic called fusidic acid (FA) was also incorporated into the hydrogel network, where a combined therapeutic effect was observed. PMB is a positively charged cyclic pentapeptide which acts by binding the negatively charged groups on the membrane of the Gram-negative bacteria and then penetrating the cell, leading to its death [70]. Peptides are known to possess groups that are labile towards light, enzymes, thermals, and different chemical conditions. Hence, they can be utilized for noncovalent (physical) linkages in various formulations. The authors utilized PMB as a cationic crosslinker and a trigger for the supramolecular gelation of peptides with oppositely charged amphiphile peptides. Three different amphiphile peptides with various amino acids were investigated for the possible PMB-triggered gelation of peptide assemblies (Figure 12) [70].

All peptides contain Glu, which possess negative charge. Furthermore, the first peptide, TPA, contains Glu-Thr-Glu to form the paired interaction with the diaminobutyric acid–threonine–diaminobutyric acid of the linear tail of the PMB. To elucidate the role of the H-bonding between the Thr residue, in the second peptide (EPA), the Thr reside was swapped with Glu. The third peptide (BPA) was designed so it can form a nanofiber structure where the amino acid series Val-Val-Val-Ala-Ala-Ala is between the hydrophobic alkyl tail and the hydrophilic head of the second peptide. CD and TEM confirmed the nanofiber formation between TPA and BPA, and for BPA, a canonical β-sheet conformation was observed (increased twisting). Mixing solutions of PMB with TPA formed the hydrogel, but this was not the case with the BPA under the same conditions. This was ascribed to the difference in the molecular arrangements between both peptides. The loose molecular packing inside the lamellar of the TPA fibers allows the PMB to be easily introduced within the assemblies. Given the favored hydrophobic interactions and the H-bonding between the hydrophobic tail and the sidechain OH group of Thr reside in TPA and PMB, which leads to the enhancement and stabilization of the weak ionic interaction between the carboxyl groups of TPA and the amino groups of the PMB, and accordingly the enhancement of the cohesion of the assemblies, further electrostatic interactions between the PMB and other TPA would promote crosslinking and hence hydrogel formation. On the other hand, the BPA, which has β-sheet configuration, possesses a strong H-bond interaction within similar molecules, resulting in cylindrical and tightly packed nanofibers which hinder the insertion of the PMB with those tightly packed nanofibers. Thus, the interaction between PMB and BPA is only electrostatic at the surface. The absence of other interactions, such as the H-bonding, led to unstable membranes, which showed in this study as only opaque thin membrane that vanished after incubation overnight at 37 °C. Interestingly, the addition of calcium ions, where the PMB was dissolved in calcium chloride instead of pure water, led to enhanced mechanical stability and stiffness of the triggered hydrogel, which is beneficial for various applications [70]. Sustained released over a five-day period was observed with either the PMB in pure water or in calcium chloride, with less release observed with the case of the latter. Interestingly, this could be exploited to optimize the dosing and adapt it to various clinical needs. The efficacy of the PMB-triggered PA hydrogels was assessed against the PMB solution using the disk transfer and diffusion method and the same final concentrations of PMB. The hydrogel formulations were able to inhibit the Gram-negative *Pseudomonas aeruginosa* PA14 with a minimum inhibitory concentration (MIC) of 0.5 µg mL^−1^, whereas the PMB solution showed no inhibitory effect at concentration as high as 20 µg mL^−1^. Interestingly, after 12 h of the disk transfer and diffusion experiment, a gradual decrease in the inhibitory effect against PA14 was observed with the PMB solution. On the other hand, with the PMB-triggered hydrogels, an Increased inhibitory effect was noticed after the first 4.5 h and which lasted to 24 h thereafter. These findings proved the sustained released pattern of those hydrogels as well as their prolonged antimicrobial activity [70]. The authors also validated the in vivo antimicrobial activity of the PMB-triggered PA hydrogels on a *G. mellonella* burn wound infection model. Interestingly, both formulations (with and without calcium chloride) performed well in reducing the mortality, with a better performance for the PMB + Ca gel over that without the Ca ion. Surprisingly, the Ca gel formulation, which is the negative control, also promoted survival, and the authors ascribed it to the hydration effect from the hydrogel treatment [70].

The authors investigated the possibility of delivering a combined effect within the hydrogel formulation. As the PMB is effective against the Gram-negative bacteria, they envisaged incorporating another antibiotic that is effective against the Gram-positive bacteria. Thus, they incorporated FA and PMB together. This gel indicates that the PMB-triggered hydrogel is not compromised by the addition of the FA. Interestingly, FA was effectively released from the FA-PMB gel and worked efficiently against Gram-positive *Staphylococcus aureus*. The combined formulation showed higher potency than the FA only after 8 h in the disk and transfer diffusion test. The authors investigated the PMB gel and FA + PMB gel against *A. baumannii* and they were both effective over a 12 h period. However, the FA + PMB gel showed larger inhibition zones than the PMB gel alone. Moreover, the soluble solution of FA + PMB had more antimicrobial activity than the FA solution in the disk transfer and diffusion test. Furthermore, the FA + PMB solution showed the antimicrobial activity after 6 h with respect to the PMB solution [70].

Cui et al. developed an instant self-assembly peptide with alginate that targets infected wounds [71]. The authors selected a short peptide that can self-assemble outside the microfibrous alginate in a weak acidic environment of a pH of about 6 in only 5 s. The authors considered a dual drug-delivery system. They constructed a recombinant bovine basic growth factor (FGF-2) in a fibrous alginate that was encapsulated in a peptide hydrogel (already preloaded with antibiotic). The idea of the dual-drug delivery system here is to release the antibiotic quickly while sustaining the release of the growth factor. Alginate was selected as a good candidate for this purpose because it is biocompatible, possesses a slow degradation profile in the pH close to that of the human skin, and it has a rapid gelling property using the microfluidic technique [71]. The peptides were selected to enhance the self-assembly (Nap-GFF), in which the π–π interactions between the aromatic phenyl of the Phe reside and that of the Nap drive the self-assembly process. To enhance the solubility of the selected peptides in water, hydrophilic residues were also incorporated, Lys and His. These four peptides were used: Nap-GFFKQHH, Nap-GFFKEH, Nap-GFFKGHH, and Nap-GFFKH. FTIR demonstrated that loading the FGF in the alginate fiber did not interfere with the properties of alginate after the encapsulation process. The degradation study in phosphate buffer solution pH 7 at 37 °C showed that the pure fiber degrades first followed by the pure peptide hydrogel (both in 1 day). The fiber–peptide hydrogel was the slowest and it degraded to a lesser extent and over a longer period (7 days) [71]. Different ratios of the alginate and the peptide were chosen, and 1:3 was selected as the optimum one in terms of degradation. After that, ampicillin and lincomycin antibiotics were loaded into the peptide hydrogel. Then, the peptide hydrogel was utilized to encapsulate the growth-loaded microfiber. The release of ampicillin@peptide and ampicillin@FGF-peptide was faster than that of FGF@fiber-ampicillin@peptide. The same observation was seen for FGF. The previous findings were also observed when utilizing the lincomycin antibiotic [71].

In summary, the new delivery system allows for sustained released for the FGF from the microfiber, as well as a burst release for the antibiotic, which is favorable in wound-healing cases. This success is ascribed to the fact that the antibiotics were physically incorporated into the peptide hydrogel, so as the peptide swells, this facilitates the burst release of the antibiotic. The in vitro study showed good inhibition effect on both bacteria *E-coli* and *S.Aureus*. Lastly, the in vivo study proved the superiority of the dual-drug to single-drug delivery approach [71].

Cao and coworkers investigated a peptide PAF26 (Ac-RKKWFW-NH_2_) that could self-assemble upon contact with the pathogenic microbes, destroying their cell walls and killing them [72]. The peptide was dissolved in water and the pH was adjusted until the hydrogel was formed, mainly by intermolecular forces and the hydrophobic effect. UV-Vis spectra showed that high concentration of PAF26 could interact via aromatic stacking (π–π) interactions. The same red shift was observed in the self-assembled hydrogel [72]. Its antimicrobial activity was investigated against *Candida albicans*, *Staphylococcus aureus*, and *Escherichia coli*. The observed inhibition zones demonstrated that the hydrogel could inhibit the pathogens and microbes even outside the hydrogel itself. The optical density at 600 nm (OD_600_) showed that the hydrogel could inhibit the pathogens’ growth. Moreover, no microbial colonies were observed, which reflects the killing efficacy of the hydrogel. The authors suggested that the destruction of the cell membrane could be the mechanism of the antimicrobial activity of the hydrogel. Scanning electron microscope (SEM) data confirmed this hypothesis [72].

In a related study done by Cao et al., a covalently crosslinked antimicrobial hydrogel via a disulfide bridge of the *C*-terminal Cys residue was created [73]. The PAF26 analog from their previous study [72] was considered in this study, in which the following peptide was considered: Ac-RKKWFWC-NH_2_. The hydrogel can be formed at a neutral pH, where the thiol group of the Cys could be oxidized by air. This hydrogel proved its ability to inhibit the growth of *Staphylococcus aureus* and *Escherichia coli* [73].

Since hydrogels formed by physical interactions are vulnerable to in vivo degradation and low mechanical stability, the authors proposed a covalently crosslinked hydrogel. After dissolving the peptide in water, the pH was adjusted by base until the hydrogel RC7 was formed. This hydrogel was quickly frozen with liquid N_2_ and lyophilized. Afterward, IR and Raman spectra confirmed that the thiol groups were oxidized and formed the crosslinks. The amphipathic structure of the peptide enhances the potential self-assembly by hydrophobicity and π-π stacking interactions. Moreover, the Cys residues will deliver the covalent crosslinking by forming the disulfide bridge upon oxidation. Antimicrobial activity study on *Candida albicans*, *Staphylococcus aureus*, and *Escherichia coli* proved the ability of RC7 to diffuse outside the hydrogel and inhibit the microbial growth of the microbes. OD_600_ experiments proved the ability of RC7 to inhibit the bacterial growth of these three bacteria, though it could only completely kill *Candida albicans* [73].

Chen and coworkers designed an injectable peptide hydrogel that can sustain the release the salvianolic acid B (SaB) for myocardial infraction [74]. SaB suffers from decomposition during the long-term treatment of myocardial infarction. Therefore, it is usually crosslinked and loaded on polydopamine (PDA) in large doses. In their work, extra precautions were taken because the continuous beating of the heart as well as the tissue compression can damage the hydrogel irreversibly [74]. Thus, a novel peptide was constructed with a hydrogel that possesses elasticity, hence it can tolerate the stretching and shrinking forces associated with the heart beating [74]. The following peptide (Fmoc-FFVPGVGQGK) represents the elastin–mimic hydrogels (EMHs) and comprises three blocks: self-assembly (Fmoc-FF), framework (VPGVG), and the transglutaminase (TGase) crosslinking (QGK). VPGVG is the abundant repeated unit of the elastin in which the presence of the Fmoc-FF drives the formation of the fiber and then the hydrogel (pre-EMH). The addition of SaB-PDA to the pre-EMH enhances the physical crosslinking. Sab-PDA/EMH is formed as a result of TGase, which is upregulated in the heart tissue postmyocardial infarction. It is considered mechanically stable and it tolerates the frequent stretching and shrinking forces, thus extending the healing time and avoiding the burst release of the drug [74]. Three different sequences were synthesized and tested for their ability to form the hydrogel: P1: Fmoc-FFVPGVGQGK, P2: QGKVPGVGQGK, and P3: Fmoc-FFVQGK. TEM analysis showed that only P1 and P2 formed the nanofibers, with P1 of >1 µm and P2 ~300 nm. The presence of Fmoc-FF drove the formation of the nanofibers while the VPGVG fragment led to larger fibers in P1. As P1 was the only peptide that formed a hydrogel, it was chosen for the main study. P1 aggregated into micelles above the critical aggregation concentration and then the micelles turned into nanofibers [74]. CD analysis showed a β-sheet secondary structure configuration. The red-shift of the aromatic groups supported that the π–π stacking interactions drive the self-assembly process. TGase enzymes catalyze the short peptides to form nanoparticles. Thus, in this study, after the addition of the TGase, new substances appeared as a result of acyl transfer reaction from Gln to Lys, which is required for the crosslinking reaction. Additional mechanical stability in this formulation is a result of the hydrogen bonds formed between the catechol groups of the PDA and the benzene rings of SaB-PDA. The optimum concentration of SaB-PDA to the pre-EMH is 2 wt%. The pH of the hydrogel solvent was 7.4, which will develop a negative charge on the PDA and lead to electrostatic repulsion with the SaB, triggering its slow release. The release after 96 h of the PDA nanoparticles without the hydrogel was 83%, for SaB/EMH without the PDA nanoparticles it was 40%, whereas for SaB-PDA/EMH it was 17%. The safety profile of the developed hydrogel was tolerable. The cell viabilities of all the groups recruited in the study were close from that of the control group. Furthermore, in vivo studies showed no cytotoxicity from SaB-PDA/pre-EMH in comparison with a saline treatment [74].

## 6. Stimuli-Responsive Peptides

Lee and coworkers developed a stimuli-responsive conformational transformation peptide in which the cytotoxicity could be controlled in response to a specific biological environment [75]. The authors considered a peptide from the honeybee venom which has a biological activity (KLAKLAK)_2_ (Figure 13A).

Besides the anticancer activity of this peptide, it can also interact electrostatically with the negatively charged phospholipids of the membranes due to the 6 Lys residues within its sequence. Thus, it is also considered as a membrane-active peptide. The hydrophobic interaction between the hydrophobic alkyl chain of the membrane phospholipids and the Leu amino acid stabilizes the α-helical conformation of the peptide [75]. The peptide in its helical form can form pores on the cell membrane and induce cell death. The KLA peptide endocytoses in the cancerous cells was eight times more than in the normal cells. To enhance the therapeutic efficacy of this peptide, the authors considered the stimuli-responsive conformational transformation. Two Cys residues were incorporated at both termini of the peptide to cyclize the peptide via these disulfide bridges (Figure 13B). The cyclized peptide possessed a constrained helicity and hence a lower cytotoxicity. Afterwards, the modified KLA-SS peptide was conjugated with hyaluronic acid (HA) by electrostatic interaction, enhancing the endocytosis into the cancer cells that express CD44. The HA is degraded by the upregulated hyaluronidases, the KLA-SS peptide is released into the cytoplasm of the cancerous cell, and the disulfide bond is reduced by the upregulated GSH into two thiol groups. Finally, the strain that suppresses the helicity is released and the original helicity is restored [75].

According to the helicity data, the authors proved that the reduced KLA-SS peptide has a comparable helicity with the parent KLA peptide (40%), but it was lower in the case of the KLA-SS before the reduction using tris(2-carboxyethyl)phosphine (TCEP) (27%). Furthermore, after electrostatic conjugation with HA, the helicity of KLA-SS/HA was 24% before the reduction and it went up to 30% after the reduction of the disulfide to thiol groups. Liposome leakage experiment also confirmed that the KLA-SS peptide with low helicity (where the experiment was done without GSH) exhibited low capability to disrupt the calcein-loaded lipid membrane. On the other hand, in the presence of GSH, the disruption capability was higher, which confirms the induced reduction of the disulfide bridge by GSH [75]. In conclusion, the cytotoxicity of the KLA peptide could be controlled and manipulated with high specificity towards the cancerous cells. The liposome assay was also conducted on the KAL-SS/HA complex with and without GSH. In both cases, the low disruption capability of the calcein–lipid membrane was observed, whereby the authors ascribed it to the low helicity of this complex. Because HAYLs were reported to degrade HA, a liposome leakage test was also done for the KLA-SS/HA in the presence and the absence of HYAL1. The leakage of calcein was comparable to that reported in the case of KLA-SS in the presence of GSH in both cases, whereas it was very low in the absence of GSH [75]. These data confirmed the ability of HYAL1 to degrade HA. The KAL-SS that left the complex after the degradation of the HA would cause membrane disruption in the presence of GSH. This explains the selective cytotoxicity of the KAL-SS/HA complex in the cancerous cells due to the high GSH content in such cells. The GSH using buthionine sulfoximine (BSO) clearly inhibited the death of the cancerous cells, confirming the importance of the GSH. Interestingly, having the disulfide bridge also contributed to increase the resistance of the peptide to the enzymatic degradation and also constrained the structure of the peptide and facilitated the specific site binding.

Liu and coworkers designed a hydrogel utilizing a bioactive sequence while keeping the inherent activity [76]. They incorporated two biologically active peptides in this study: (KIGAKI)_3_-NH_2_ and a central tetrapeptide linker Thr-(D)Pro-Pro-Gly. As confirmed by CD experiment, the former does not self-assemble into a well-ordered structure. Thus, the latter tetrapeptide, which is also bioactive, was incorporated as a linker between two molecules of the former. This linker has a (D)Pro-Pro sequence which has the tendency to nucleate hairpins, while the Thr and the Gly were selected to occupy the i and i + 3 positions of the loop, which is inspired from the naturally occurring four amino acid β-turns. The presence of (D)Pro-Pro ensures that the flanking chains are spatially close to each other, hence facilitating the noncovalent interactions which result in a highly stable structure, as is the case for peptide 1 (ASCP1) (Figure 14A). Peptide 2 (ASCP2), which has *cis*-prolyl amide bond geometry, puts the flanking sequences away from each other (in opposite directions), and hence facilitates aggregation of the peptide (Figure 14B) [76].

By balancing the electrostatic repulsion of the Lys residues and the hydrophobic interactions of Ile and Ala residues, and in response to stimuli such as pH, heat, and ionic strength, the formation of individually dispersed nanofibers can occur. At a concentration of 5 mg/mL in an aqueous solution, ASCP1 is able to self-assemble, producing a transparent hydrogel. The predominant β-sheet was confirmed through Congo red testing, which is responsible for fiber network formation. At a concentration of 0.1 mg/mL, ASPC1 also showed a predominant presence of β-sheet at pH 11, also confirmed by CD, where a sharp minimum at 216 nm was observed. Moreover, FTIR was also carried out, and a strong absorption at 1630 cm^−1^ was observed which reflected the parallel orientation of the hydrogen bonding to the long axis of the nanofiber. The absorption at 1680 cm^−1^ suggests the antiparallel alignment of the peptide strands [76]. In summary, the ASCP1 flanking sequences adopt an antiparallel β-sheet that comprises a parallel hydrogen bonding with the nanofiber. At a concentration of 5 mg/mL, and once the pH is adjusted to 11, the hydrogel is formed within few minutes. The storage modulus (G′) kept increasing with time, reflecting the continuous growth of the gel, whereas the loss modulus (G″) remained constant with time, proving that the peptide behaved as an elastic gel. On the contrary, ASCP2 showed an irregular secondary structure even at a pH as high as 11 and remained as liquid with G″ > G′. ASCP1 was able to reach a maximum fiber length of 2–3 µm within 8 min (upon exposing to high borate ions of pH 11). The authors concluded that the minimum concentration of ASCP1 to form a β-sheet structure is 0.012 mg/mL, while, when going below this concentration, a random-coil conformation is observed instead. They also concluded the importance of the pH, in which ASCP1 was not able to self-assemble in the pH range of 2–8.8 and behaves instead as a liquid. Subsequently, as Lys has a pKa of 10.5 of its sidechain, this means that the peptide will have 12 positive charges, therefore preventing the flanking sequences from getting in close proximity due to the electrostatic repulsion, whereas, at a pH of more than 10, the lysine sidechain will have no charge and the peptide will start to collapse and aggregate. The addition of salt, such as NaCl, enhanced the formation of the β-sheet structure of ASCP1 peptide, which could be ascribed to the presence of the high density of hydrogen bonding. Moreover, the G′ has also increased in the presence of NaCl and the hydrogel rigidity correlated well with the concentration of the added NaCl. ASCP2 exhibited only a random-coil structure, even after the addition of NaCl. The effect of the temperature on the self-assembling was shown to be important. In the case of ASCP1, the formation of the β-sheet structure was enhanced in the temperature range from 60 to 70 °C, whereas at 50 °C, the random-coil structure was predominant. Interestingly, the G′ values were also enhanced from 13 to 20 Pa at 20 °C to 4000 Pa at 65 °C. Most importantly, this phase transition behavior was thermally reversible. Again, in the case of ASCP2 peptide, it showed no aggregation with the increasing the temperature. The interpretation is based on the balance between the hydrophilicity and hydrophobicity, which is altered as a result of the temperature-related changes in the solubility. Finally, the biological activity of ASCP1 was mainly ascribed to the positive charges of this peptide at physiological pH, which attracted the negatively charged bacterial membrane and led to its disruption and eventually cell death [76].

Hartlieb and coworkers designed membrane-active cyclic peptide nanotubes (CPNT) in which their activity was masked with a conjugate polymer until they reached their targets. Then, the macromolecule could be cleaved in response to certain stimuli, hence restoring the CPNT activity [77]. Interestingly, during the activation process, it was possible to control the nanotubes with an increased length instead of aggregates. In this work, the peptide was selected to be with an even number of amino acids with alternating chirality. The authors tried to overcome obstacles such as the aggregation of nanotubes and a lack of membrane specificity by combining the CPNT with polymers through covalent conjugation [77]. They selected a reversible covalent structure which could be cleaved upon changing the solution conditions. The polymer forms a shell that contains the CPNT, hence preventing any undesired interactions with the membranes and any other surfaces. At the target site, the polymer will be cleaved and the CPNT will be active again. In this study, the authors choose Poly(2-ethyl-2-oxazoline) (PEtOx) as a hydrophilic and biocompatible polymer. It was linked to the peptide through a cleavable disulfide linker Cyclo(L-Trp–D-Leu–L-Lys–D-Leu–L-Trp–D-Leu–L-Lys–D-Leu) (Figure 15) [77].

The hydrophilic polymeric shell keeps the CPNT inactive until it reaches its target. Then, an on-demand detachment of the polymer takes place in response to certain chemicals that are able to cleave the disulfide linker. The polymer was anchored through the sidechain of the Lys residue (Figure 15). Different conjugation approaches can lead to either responsive or nonresponsive materials [77].

The authors studied different polymerization degrees of PEtOx (10, 20, and 45) to investigate the influence on the self-assembly behavior as well as the responsive properties. PEtOx_45_ showed aggregation with no differences between the responsive and nonresponsive systems. A higher degree of aggregation was observed with PEtOx_20_, which ascribed to the lower hinderance imposed by the polymer. A significant difference was observed between both responsive and nonresponsive systems: 3.5-fold more in the case of responsive system. The detachment was achieved by reducing the sulfide linker with the aid of 30 mM of 1,4-dithiothreitol (DTT). The size of the polymer can also play an important role in stabilizing the nanotubes against aggregation, given that a thicker polymer shell could shield the cleavable sulfide linker from the DTT-reducing agent, leading to reduced reaction rate. [77]. Interestingly, there is also an opportunity to recruit the natural glutathione (GSH) as an endogenous reducing agent and tune the response to this stimulus. Considering a maximum concentration of 10 mM, only PEtOx_20_ showed aggregation after 24 h, with the authors using synthetic liposomes, a mixture of phosphatidylethanolamine and phosphatidylglycerol, to mimic the cytosolic membrane of *E. coli*. They loaded the vesicles with a dye (calcein) to monitor the disruption of the bilayer through the increase in the fluorescence as a result of dye leakage. Membrane disruption was only observed upon the addition of the DTT to the system, while the nonresponsive system, as well as the responsive one (in the absence of DTT), caused no disruption to the liposomes [77]. An additional experiment was done using DTT only, where no disruption was observed, which excludes the DTT of being a factor in the disruption process. The authors concluded the minimum concentration of CPNT that causes the disruption was 0.21 µg/mL. Moreover, the lowest concentration of DTT that can accomplish the disruption was determined to be 5 mM. GSH was not able to cause a disruption in the liposome (up to 30 min), but a longer experimental period was not considered due to bleaching issues with the calcein dye. The developed system in this study was able to represent the naturally occurring membranes. The presence of the PEtOx shell shields the nanotube efficiently, in which the hemolysis level did not exceed the normal levels of 2%. The addition of the DTT caused an increase in that level in the case of the responsive system, while the nonresponsive one remained unaffected. No toxicity against the mammalian cells was observed, even with the bare cyclic peptide itself. Moreover, the addition of the DTT and cytotoxicity testing (within a short period of the addition) showed no toxicity [77].

Table 1 summarizes the various drug delivery tactics and their role in enhancing the overall delivery process.

## 7. Conclusions

Peptides are key chemicals for applications including drug discovery, immunology, diagnostic, among others. Peptides are consolidating their presence in the pharmaceutical industry, with a total of 22 peptides approved over the last six years [7]. The research in this filed is ongoing, but innovative ideas are needed to manipulate their efficacy, specificity, safety profile, the stability of these molecules, and eventually to bring them into the market.

Substantial chemical modifications are deemed important to circumvent the stability concerns of peptide-based molecules. The engineering of peptide structures is considered a promising strategy that has shown great enhancements in their stability, helicity, half-life, and biological activity. Furthermore, various innovative formulations have also helped in boosting the efficiency of peptides via protecting them from the enzymatic degradation in the first place, and also by providing different active release mechanisms, including over prolonged periods.

Interestingly, peptide molecules are also considered for use as delivery vectors, further confirming the wide applications of this valuable class of molecules. For example, in response to certain stimuli at a desired site, a peptide can be transformed from one structure to a more effective one. A variety of elegant formulation technologies have shown promising outcomes in terms of providing a protective shell for the peptide until it is delivered to its therapeutic target. Formulation tactics include on-demand detachment to free the active peptide in the desired target locations.

Remarkable advancements have been made in delivering peptides to their therapeutic targets. However, it is to be noted that, despite those advancements, some peptides are still not able to reach the market. For example, although CPPs were first introduced in 1988, the USFDA has not yet approved any CPPs either for therapy or imaging purposes or as an active drug. One reason is due to the lack of their specificity, in which, postpenetration, in addition to the intended target, they often target other neighboring cells as well [78]. Some exceptions do exist; for example, the TD2.2 peptide is reported to target oligodendrocytes and has been shown to not target nonglial cells [79]. Another reason could be ascribed to the unidentified penetration mechanism, in which each CPP can exploit various routes to penetrate the cell and sometimes more than one route can operate concurrently. This could raise safety concerns and defer their approvals. In the stapling strategy, some amino acids have to be swapped with the hydrocarbon for the stapling purposes, which is expected to compromise the biological activity of the peptide [60]. Moreover, the stapling strategy is restricted to peptides with a helical structure. Often the concepts of cyclization and stapling have been interchanged. However, stapling signifies a refinement of cyclization. It is not only to put two moieties of a linear peptide together, but also the peptide acquires a very well-defined structure. In this regard, the upcoming years will witness new stapling strategies, which will allow tailoring the structure. Furthermore, some of them bear a third functional moiety for further conjugation. Liposomes have proven to be good candidates for in vitro and in vivo delivery. However, their low entrapment efficiency as well as cargo leakage are considered as limitations [80]. Despite the huge benefits of the hydrogel to deliver peptides, they have to possess an appropriate mechanical stability which can be manipulated by altering the peptide concentration, ionic strength, and pH. However, these conditions might not be appropriate for the intended therapeutic applications [81]. 

Thus, considering the pros and cons of the available strategies, the research is still ongoing and alternatives are being investigated. One of the promising strategies is the hydrophobic ion pairing (HIP). It is used to encapsulate the charged hydrophilic molecules by forming a neutral ion pair with an oppositely charged hydrophobic moiety. HIP offers various attractive delivery possibilities, such as coencapsulating hydrophilic and hydrophobic therapeutic drugs, regulating the drug release mechanism, forming the HIP ion pair from two therapeutic entities, among others. Interested readers are encouraged to refer to the following review [82].

The field is being revolutionized to include modern strategies such as synthetic nanochaperone [83] and halloysite nanotubes [84]. In conclusion, the field is in need for novel ideas which could help introduce these peptides to the market.

## Figures and Tables

**Figure 1 pharmaceuticals-15-01283-f001:**
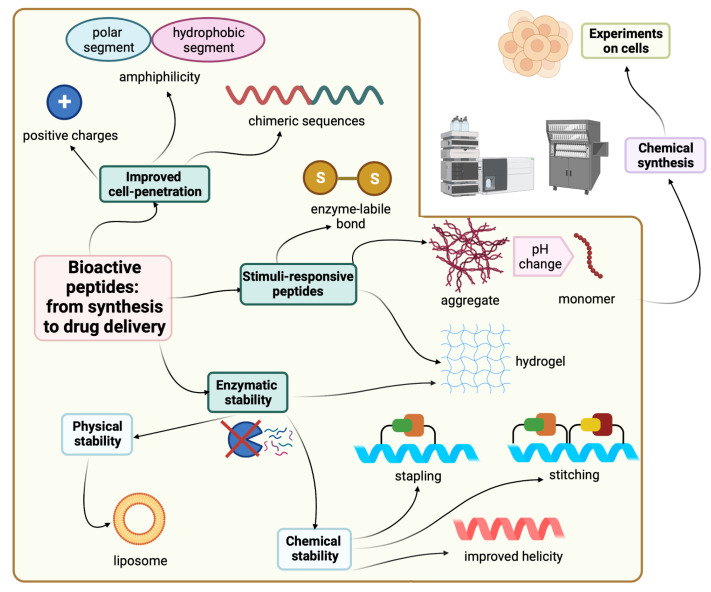
Strategies and tactics to improve peptide stability and delivery. Created with Biorender.com.

**Figure 2 pharmaceuticals-15-01283-f002:**
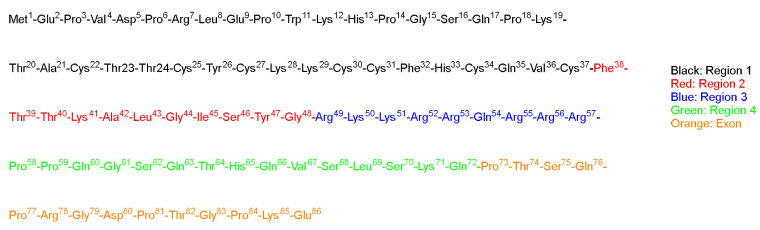
TAT sequence showing the tentative functional regions (adopted from [26]).

**Figure 3 pharmaceuticals-15-01283-f003:**
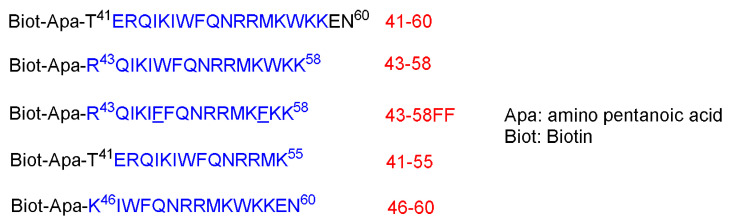
Different sequences. The region that corresponds to the third helix of the homeodomain is shown in blue. The two new F residues that replaced the W are underlined.

**Figure 4 pharmaceuticals-15-01283-f004:**
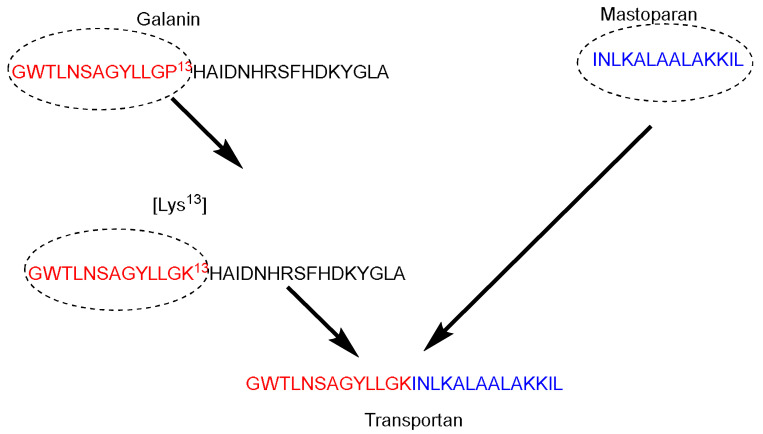
Transportan chimeric peptide sequence. Red: the sequence from galanin neuropeptide; Blue the sequence from the wasp venom peptide toxin (mastoparan).

**Figure 5 pharmaceuticals-15-01283-f005:**
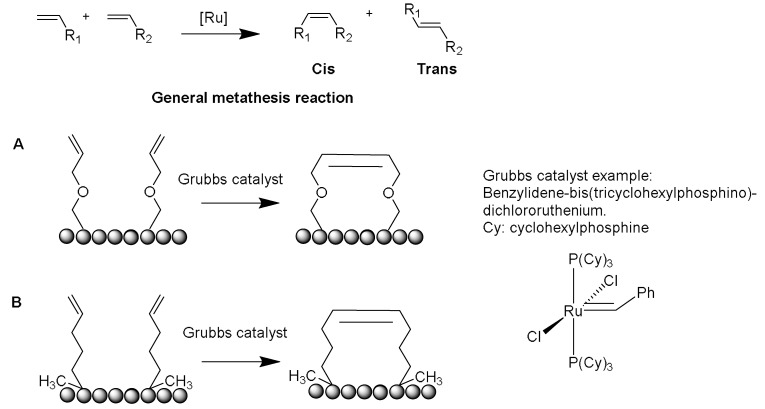
Ruthenium catalyzed olefin metathesis (RCM). (**A**) *O*-allyl serine residues by Grubbs [59]. (**B**) The α,α-disubstitution, olefin tether by Verdine [60].

**Figure 6 pharmaceuticals-15-01283-f006:**
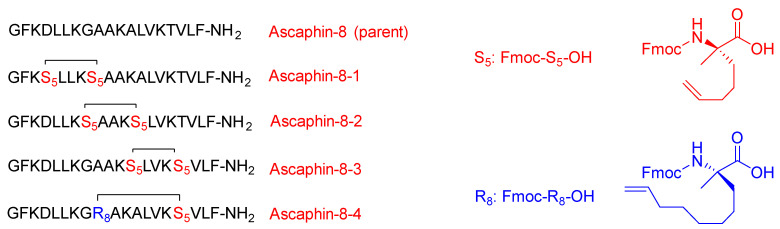
The sequences of the parent Ascaphin-8 peptide and four analogs with different stapling positions. Fmoc-S_5_-OH (Fmoc-(S)-2-(4-pentenyl)Ala-OH) and Fmoc-R_8_-OH (Fmoc-(R)-2-(7-octenyl)Ala-OH) are shown at the right in red and blue, respectively.

**Figure 7 pharmaceuticals-15-01283-f007:**
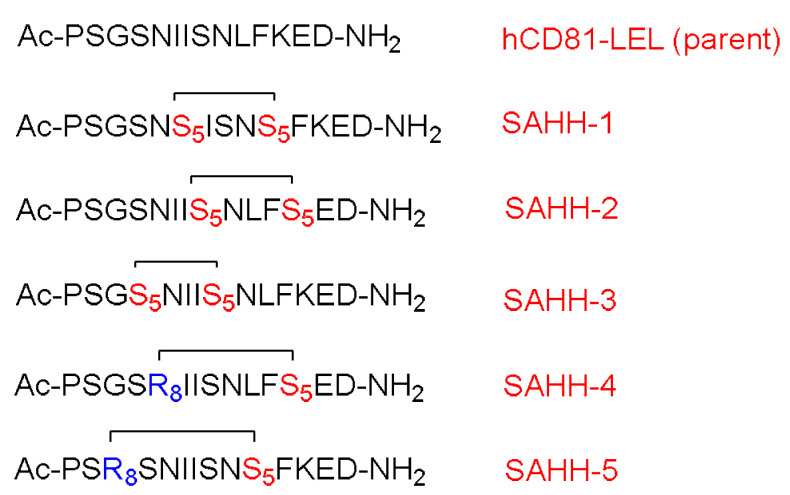
The sequences of the parent hCD81-LEL peptide and five analogs with different stapling positions. Fmoc-S_5_-OH and Fmoc-R_8_-OH are shown at the right in red and blue, respectively. The parent peptide was identified and synthesized by Dhanasekaran et al. [62]. For S_5_ and R_8_, refer to the legend of Figure 6.

**Figure 8 pharmaceuticals-15-01283-f008:**
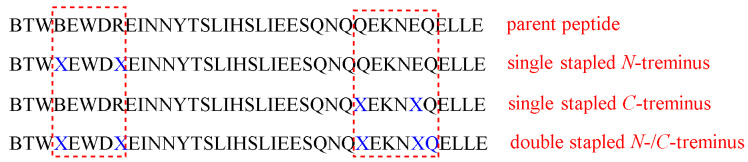
T649v parent peptides and the corresponding stapled modified analogs. X in blue represents the stapling positions.

**Figure 9 pharmaceuticals-15-01283-f009:**
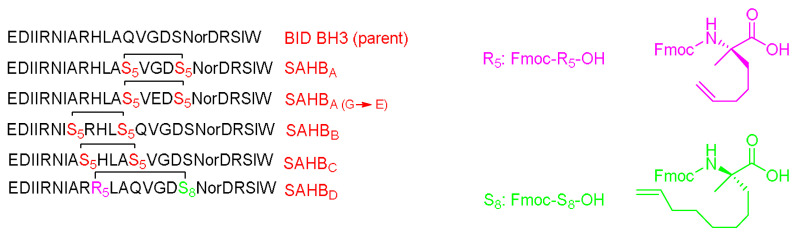
BID-BH3 parent peptide and the corresponding five analogs with different stapling positions. Nor: Norleucine. Fmoc-R_5_-OH (Fmoc-(R)-2-(4-pentenyl)Ala-OH) and Fmoc-S_8_-OH (Fmoc-(S)-2-(7-octenyl)Ala-OH) are shown at the right in pink and green, respectively. For S_5_ refer to the legend of Figure 6.

**Figure 10 pharmaceuticals-15-01283-f010:**
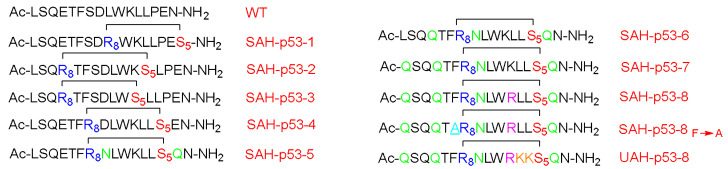
SAH-p53 parent peptide and ten corresponding analogs with different stapling positions. Both Asn and Gln that replaced Asp and Glu residues are shown in green; Arg that replaced Lys is shown in pink; Lys that replaced Leu is shown in orange. For S_5_ refer to the legend of Figure 6.

**Figure 11 pharmaceuticals-15-01283-f011:**
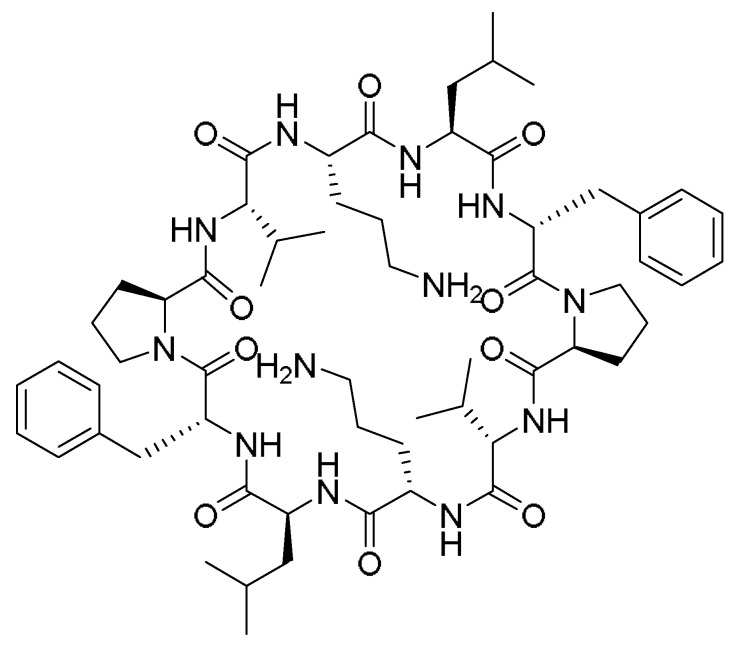
Chemical structure of Gramicidin S (GraS).

**Figure 12 pharmaceuticals-15-01283-f012:**
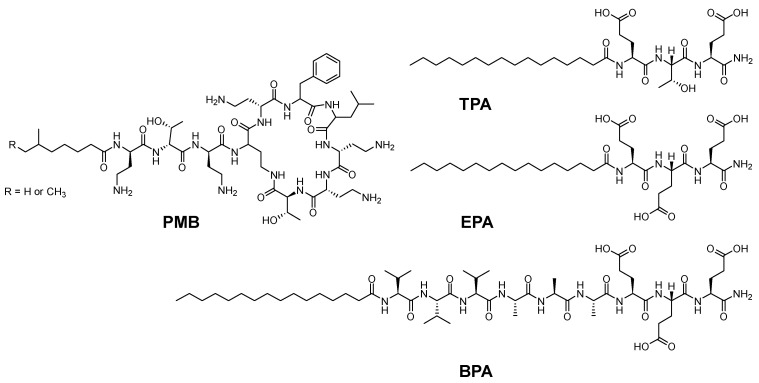
Chemical structures of polymyxin B (PMB) and the three amphiphile peptides (PA): threonine (T)-containing PA (TPA), glutamic acid (E)-containing PA (EPA), and beta-sheet-forming PA (BPA).

**Figure 13 pharmaceuticals-15-01283-f013:**
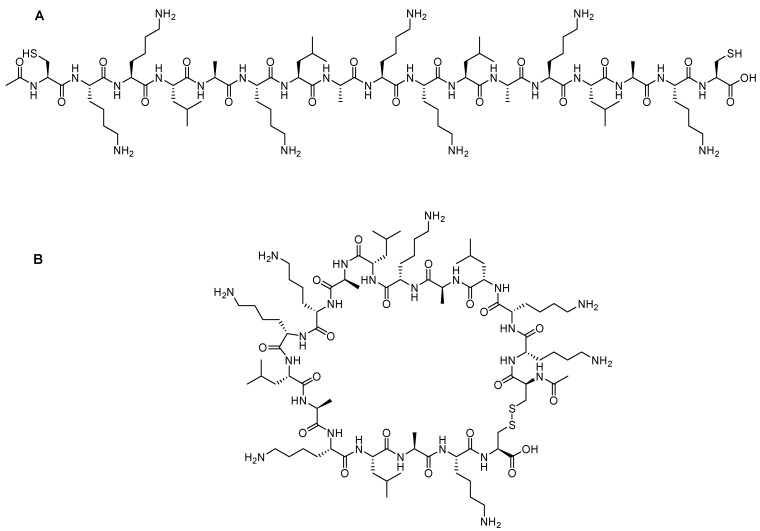
Chemical structure of (**A**) Linear KLA-SS peptide (**B**); Cyclic KLA-SS peptide.

**Figure 14 pharmaceuticals-15-01283-f014:**
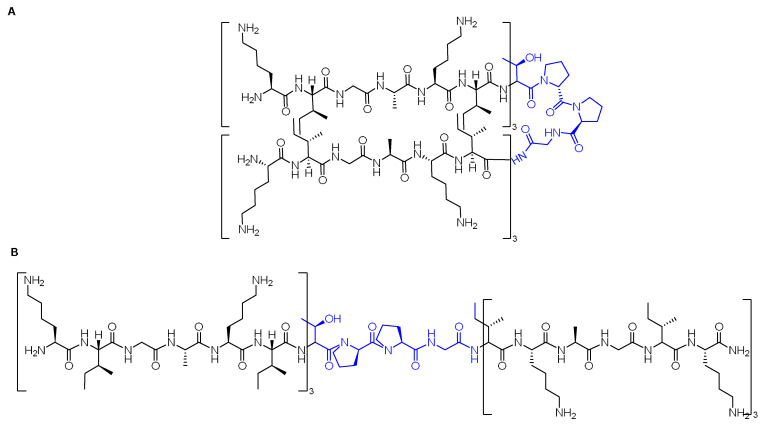
Chemical structure of: (**A**) ASCP1; (**B**) ASCP2. The linker is shown in blue.

**Figure 15 pharmaceuticals-15-01283-f015:**
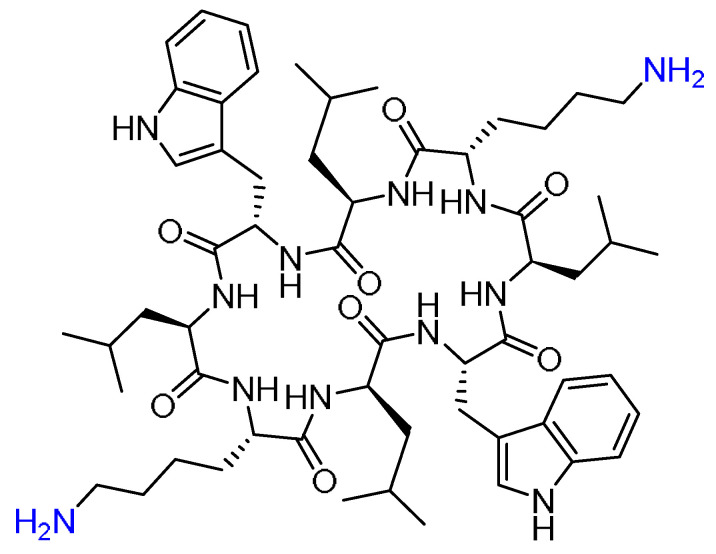
Chemical structure of the cyclic peptide comprising alternating L and D amino acids. Anchoring points are shown in blue.

**Table 1 pharmaceuticals-15-01283-t001:** Summary of the peptides and the modifications to overcome delivery challenges.

Peptide	Modification	Effect of the Modification	Ref.
Ascaphin-8	Stapling i, i + 4 & i, i + 7	Higher stability Higher toxicity	[61]
PSGSNIISNLFKED and GSSTLTALTTSVLKNNL	Stapling i, i + 4 & i, i + 7	Higher helicity Better inhibitory effect of HCV	[62,63]
Enfuvirtide	Single and double stapling [two pairs of (i, i + 4)]	Enhanced half-life Higher proteolytic resistance	[64]
BH3 peptide	Stapling i, i + 4 & i, i + 7	Higher helicity Higher protease resistance Higher cell penetration ability higher binding affinity towards BCL-2	[65]
p53	Stapling i, i + 7	Enhanced affinity towards hDM2 Higher cell penetration ability	[66]
Oligoarginine R12	Stapling i, i + 7 Stitching i + 4 + 7	Higher helicity Higher cell penetration ability High thermal stability, even higher in the stitched analog Higher protease resistance, even higher in the stitched analog	[67]
Gramicidin S (Gras)	Liposome formulation	Higher cell penetration ability	[68]
H-EKKRLLKWWR-NH_2_ and H-KWWKLLRKKR- NH_2_	Liposome formulation	Enhanced permeability	[69]
Polymyxin B (PMB)	Hydrogel formulation	Enhanced inhibitory effect	[70]
Nap-GFFKQHH Nap-GFFKEH Nap-GFFKGHH Nap-GFFKH	Hydrogel formulation	Allowed sustained release Enhanced inhibitory effect	[71]
PAF26 (Ac-RKKWFW-NH_2_)	Hydrogel formulation	Killing efficacy of the hydrogel itself	[72,73]
Salvianolic acid B (SaB)	Hydrogel formulation	Allowed sustained release	[74]
(KLAKLAK)_2_	Stimuli-responsive formulation	Higher helicity Higher membrane disruption capability Selective toxicity	[75]
(KIGAKI)_3_-NH_2_	Stimuli-responsive formulation	Higher membrane disruption capability	[76]
Cyclo(L-Trp–D-Leu–L-Lys–D-Leu–L-Trp–D-Leu–L-Lys–D-Leu)	Stimuli-responsive formulation	Higher specificity Higher protease resistance	[77]

## Data Availability

Data sharing not applicable.

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
