# Peer review of "Strategies for Improving Peptide Stability and Delivery"

_pharmaceuticals, 2022, doi:10.3390/ph15101283_

Round 1

Reviewer 1 Report

This is an interesting and well-organized paper. I strongly recommend the publication of this paper. Before publication, I suggest the authors could provide some figures (the cartoon will be better, and some important results) to attract more readers, such as in 1) 2. Cell-penetrating peptides; 2) Stapled peptides; 4) Liposomes; 5) Hydrogels. In addition, the future outlook should be provided. 

Author Response

Our reply has been uploaded. Thanks for your input!

Reviewer 2 Report

The manuscript by Musaimi et al. ‘Strategies for improving peptide stability and delivery’ seems interesting and informative. The overall manuscript looks good. However, the following points should be considered before publication.

1.    Authors have discussed several strategies for improving peptide stability and delivery such as cell-penetrating peptides, stapled peptides, liposomes, hydrogels, and stimuli-responsive peptides. Recently, the hydrophobic ion-pairing (HIP) strategy has been widely utilized for enhancing peptide stability and delivery. It will be better if authors also discuss widely utilized strategies, including HIP and so on.

2.    Also, discuss the limitations of existing strategies for improving peptide stability and delivery

3.    Include future perspective and research direction 

Author Response

(The authors gave the same response as above.)
